# TASOR epigenetic repressor cooperates with a CNOT1 RNA degradation pathway to repress HIV

Roy Matkovic [1]✉, Marina Morel[1], Sophie Lanciano [2,4], Pauline Larrous[1,4], Benjamin Martin[1,4], Fabienne Bejjani[1], Virginie Vauthier[1], Maike M. K. Hansen [3], Stéphane Emiliani[1], Gael Cristofari [2], Sarah Gallois-Montbrun[1] & Florence Margottin-Goguet [1]✉

The Human Silencing Hub (HUSH) complex constituted of TASOR, MPP8 and Periphilin recruits the histone methyl-transferase SETDB1 to spread H3K9me3 repressive marks across genes and transgenes in an integration site-dependent manner. The deposition of these repressive marks leads to heterochromatin formation and inhibits gene expression, but the underlying mechanism is not fully understood. Here, we show that TASOR silencing or HIV-2 Vpx expression, which induces TASOR degradation, increases the accumulation of transcripts derived from the HIV-1 LTR promoter at a post-transcriptional level. Furthermore, using a yeast 2-hybrid screen, we identify new TASOR partners involved in RNA metabolism including the RNA deadenylase CCR4-NOT complex scaffold CNOT1. TASOR and CNOT1 synergistically repress HIV expression from its LTR. Similar to the RNA-induced transcriptional silencing complex found in fission yeast, we show that TASOR interacts with the RNA exosome and RNA Polymerase II, predominantly under its elongating state. Finally, we show that TASOR facilitates the association of RNA degradation proteins with RNA polymerase II and is detected at transcriptional centers. Altogether, we propose that HUSH operates at the transcriptional and post-transcriptional levels to repress HIV proviral expression.

[1] Université de Paris, Institut Cochin, INSERM, CNRS, 75014 Paris, France. [2] Université Côte d'Azur, Inserm, CNRS, IRCAN, Nice, France. [3] Institute for Molecules and Materials, Radboud University, 6525 AM Nijmegen, The Netherlands. [4] These authors contributed equally: Sophie Lanciano, Pauline Larrous, Benjamin Martin. ✉email: roy.matkovic@inserm.fr; florence.margottin-goguet@inserm.fr

As of today, highly active antiretroviral therapy is very efficient in inhibiting the replication of human immuno-deficiency virus (HIV) in CD4+-infected T cells, thanks to the combination of several molecules targeting different stages of the lentivirus cycle. As long as the treatment is properly followed, the viremia of the infected individual will become and remain undetectable. However, if the treatment is randomly taken off or is interrupted, a viral rebound will cause new cellular infections, increasing the number of reservoir cells in which the virus remains latent. At the proviral level, latency can be explained by an inhibition of viral transcription or by a defect in the post-transcriptional steps leading to a lack of production of new lentiviral particles. After integration, HIV-1 transcription involves cellular class II transcriptional complexes, as well as HIV proteins such as the Transactivator of transcription Tat protein, to ensure the production of genomic and subgenomic RNA species leading to viral production. However, HIV proviral expression is also dampened by multiple epigenetic mechanisms, especially when integration occurs in poorly transcribed genes or geneless regions[1–4]. The Human Silencing Hub (HUSH) complex—formed by TASOR (alias C3orf63, FAM208A), MPP8, and PPHLN-1—was identified as a potential player in HIV repression, propagating repressive H3K9me3 marks with the help of the histone methyl-transferase SETDB1 and resulting in position-effect variegation in an HIV-1 model of latency[5,6]. HUSH also contributes to silence retrotransposons such as the Long INterspersed Elements—Class 1 (L1s), preferentially those belonging to the youngest families (<5 million-year-old (myo)), both in mouse embryonic stem cells (ESCs)[7,8] and in human cells[9,10]. In addition, this complex represses hundreds of cellular genes, many of which are close to or contain retrotransposons in their introns, in particular with the help of KAP1/Trim28[8,9]. TASOR-dependent H3K9me3 deposition was also shown to occur on ribosomal DNA, ZNF encoding genes, and on repetitive or rapidly evolving genes[6,9,11,12]. Altogether HUSH appears to play a central role in maintaining genome integrity, whether the threat comes from within with L1 elements or from without with viral infections, like HIV.

We and others have previously shown that the lentiviral proteins Vpr and Vpx, and namely, Vpx from HIV-2, could counteract HUSH by preferentially inducing the degradation of TASOR and MPP8, consistent with a critical role for HUSH in antiretroviral innate immunity[5,13,14]. HUSH degradation leads to a decrease of H3K9me3 repressive marks on HIV-1 internal sequence and to the subsequent accumulation of LTR-initiated viral transcripts[5]. This increase is amplified by tumor necrosis factor-α (TNFα) treatment, a well-known inducer of transcription activation from the HIV-1 promoter[15], which led us to question a possible co- or post-transcriptional repressive effect of TASOR. In support of this hypothesis, HUSH targets were found enriched in transcriptionally permissive euchromatin[8,10]. Hence, HUSH-mediated L1 silencing often occurs in introns of transcriptionally active genes, leading to the downregulation of host gene expression[10]. In addition, TASOR and Periphilin-1 were found associated with messenger RNAs[16,17] or with the XIST long non-coding RNA (lncRNA)[18]. Here we address the question of a possible co- or post-transcriptional repressive effect of TASOR toward HIV. Our results led us to propose a model in which HUSH cooperates with CNOT1 (CCR4-NOT transcription complex subunit 1) to control viral and host gene expression through a feedback loop mechanism spanning RNA synthesis, RNA stability, and deposition of repressive epigenetic marks.

## Results

**TASOR destabilizes HIV-1 LTR-driven transcripts.** To address the question of the molecular mechanism employed by TASOR to repress HIV-1 expression, we chose a cellular system in which the LTR-driven active transcription could be studied independently of RNA splicing. Therefore, we used HeLa cells harboring one unique and monoclonal copy of an integrated LTR-Luciferase construct with a deleted TAR sequence (ΔTAR) to model a fully active transcription process[19] (Fig. 1a). Indeed, removing the TAR RNA structure, which blocks RNA polymerase II (RNAPII) elongation, leads to a 100-fold increase of Luciferase (Luc) transcript level as compared to the WT LTR in the absence of Tat (Fig. 1a). Under these experimental conditions, we confirmed that TASOR acts as a repressor of HIV-1 expression by measuring the Luc activity upon overexpression of TASOR in CRISPR/Cas9-depleted cells (Fig. 1b), or in contrast, when reducing its expression by small interfering RNA (siRNA; Fig. 1c). Notably, TASOR lacking its N-terminus PARP13-like PARP domain (ΔPARP), required for epigenetic repression according to Douse et al.[9], does not repress LTR-driven expression in contrast to full-length TASOR (Fig. 1b, up to 170 vs 47% of initial luciferase activity respectively). To examine the role of TASOR in mRNA metabolism, we performed *Nuclear Run On* experiments to assess the effect of TASOR silencing on nascent *Luc* RNA transcription from the HIV-1 LTR, while quantifying this transcript at the total level. Then, by comparing nascent *Luc* transcripts (labeled with BrUTP) to the total *Luc* mRNA amounts (Fig. S1a), we can determine at which stage of RNA metabolism TASOR may negatively act. While TASOR downregulation by siRNA very slightly increases the levels of nascent transcripts (1.3-fold, Fig. 1c), as also observed on WT LTR cells (Fig. S1b), it triggers a 2.6-fold increase of steady-state *Luc* RNA levels, suggesting that TASOR depletion also impairs the turnover of LTR-driven transcripts (Fig. 1c). Of note, the increase of luciferase activity following TASOR depletion was higher than the increase of *Luc* RNA levels when quantifying total RNA levels, suggesting that TASOR might impact steps beyond RNA degradation, such as RNA export or its translation (Fig. 1c, 9.4- vs 2.6-fold).

Next, we used HIV-2 Vpx to further confirm the role of TASOR on transcripts turnover. Indeed, mimicking TASOR siRNA treatment, Vpx expression leads to TASOR depletion (Fig. 1d, left) and to an increase of *Luc* RNA accumulation relative to its transcription, consistent with a post-transcriptional impact of TASOR on LTR-driven transcripts stability (Fig. 1d, right). By contrast, Vpx only influences nascent transcript levels of the *MORC2* gene, a known cellular target of HUSH[12] (Fig. S1c). As expected, a mutated version of Vpx (R42A), unable to induce TASOR degradation (Fig. 1d, left) but still able to induce SAMHD1 degradation (Fig. S1e), has no effect on *Luc* or on *MORC2* RNAs (Fig. 1d, right and S1c). Furthermore, HIV-1 Vpr, which presents structural similarities with Vpx but is unable to induce TASOR depletion[5,20], was also unable to increase the expression of the LTR-ΔTAR-driven transcript at any stage (Fig. 1d), although it could specifically stabilize *TNFα* transcripts (Fig. S1d), in agreement with previous observations[21]. Altogether, these results suggest that Vpx is able to induce the stabilization of the LTR-driven transcripts through TASOR degradation. Structure–function analysis provided additional hints on the potential mechanism of TASOR post-transcriptional effect. Consistent with a recent study[9], we found, using the pGenTHREADER[22] and FFPRED[23] bioinformatics tools, that the N-terminus of TASOR is predicted to fold into a PARP13-like PARP domain with high probable and reliable functions in mRNA binding, splicing and processing, together with the SPOC (Spen paralog and ortholog C-terminal) domain, often associated with transcription repression (Table S1). Moreover, bioinformatic structure prediction and amino-acid alignment of different SPOC domains showed that TASOR domain is evolutionary related to the SPOC domain of *Arabidopsis thaliana* FPA, a protein

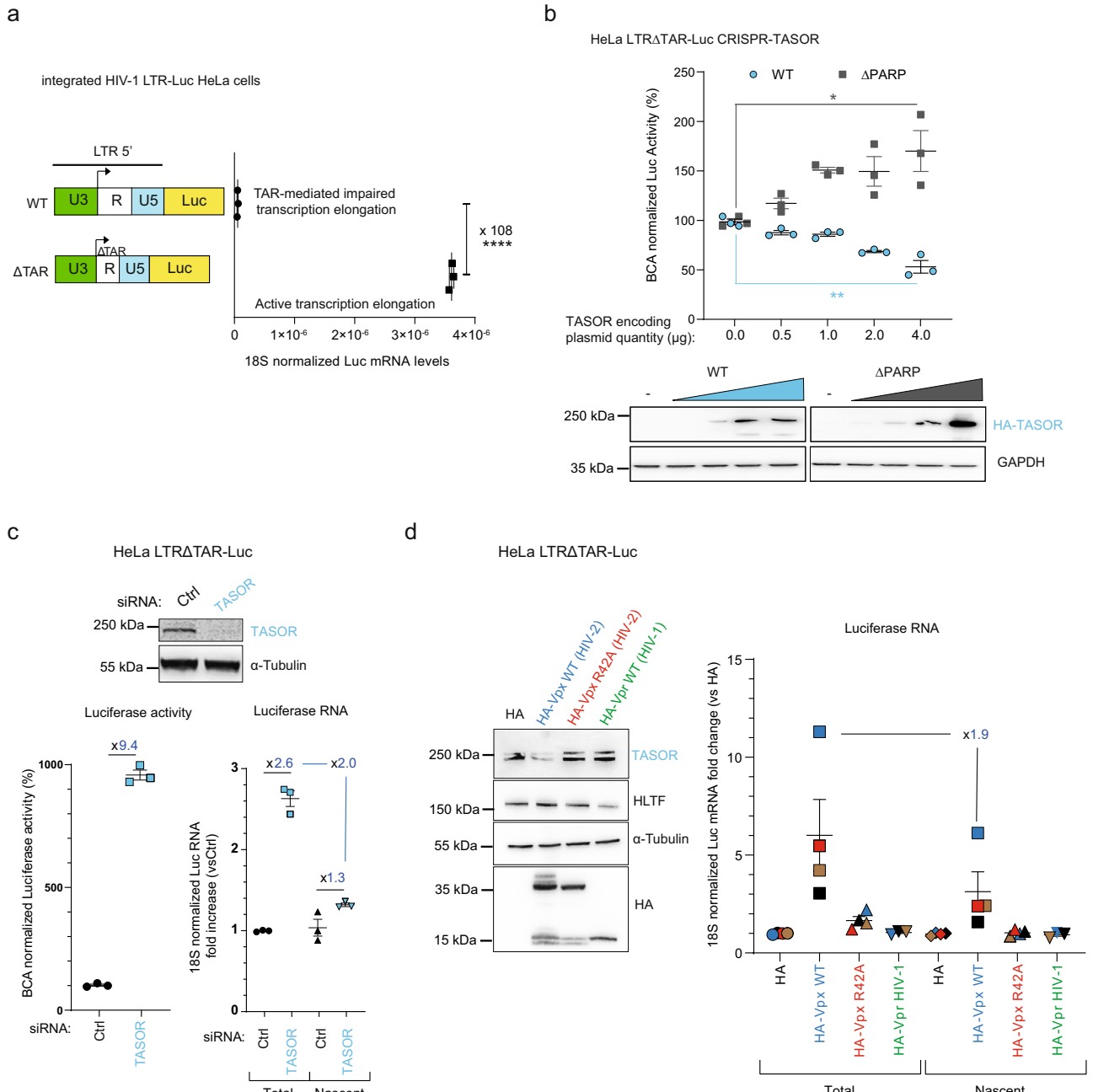

**Fig. 1 TASOR destabilizes HIV-1 LTR-driven transcripts. a** HeLa HIV-1 LTR-ΔTAR-Luc cell model. HeLa HIV-1 LTR-Luc and HeLa HIV-1 LTRΔTAR-Luc cells were lysed for Luc RNA quantification ($n = 3$; each replicate is presented along with the mean values and the SEM. A two-sided unpaired $t$ test was used. ****$p < 0.0001$). **b** TASOR overexpression decreases LTR-driven Luc expression. HeLa HIV-1 LTR-ΔTAR-Luc were TASOR-depleted by transient transfection of the pLenti-CRISPR-V2 and a sgRNA guide targeting the first exon of TASOR. These TASOR-depleted cells were then transfected with increasing amounts of HA-TASOR WT or HA-TASORΔPARP encoding pAS1B vectors. Luc activity was measured and proteins were analyzed by western blot 48 h post-transfection ($n = 3$; each independent replicate is presented along with the mean values and the SEM. A two-sided unpaired $t$ test was used. **$p = 0.0029$; *$p = 0.0262$. **c** TASOR negatively impacts LTR-driven Luc transcript at a post-transcriptional step. HeLa HIV-1 LTRΔTAR-Luc were transfected for 72 h with siCtrl or siTASOR. The luciferase activity was measured and *Nuclear Run On* experiments were performed ($n = 3$, each independent replicate is presented along with the mean values and the SEM). **d** HIV-2 Vpx mimics siRNA-mediated TASOR silencing in increasing LTR-driven transcript stability. *Nuclear Run On* performed in HeLa HIV-1 LTRΔTAR-Luc after 48 h of pAS1B-HA, pAS1B-HA-Vpx WT or R42A HIV-2 Ghana, and pAS1B-HA-Vpr HIV-1 transfection. The HIV-1 LTR-driven luciferase RNA expression was measured by RT-qPCR ($n = 4$, each independent replicate is presented along with the mean values and the SEM) and a western blot analysis monitored the levels of expression of the lentiviral proteins, TASOR, and HLTF (as an HIV-1 Vpr target[74]). Source data are provided as a Source data file.

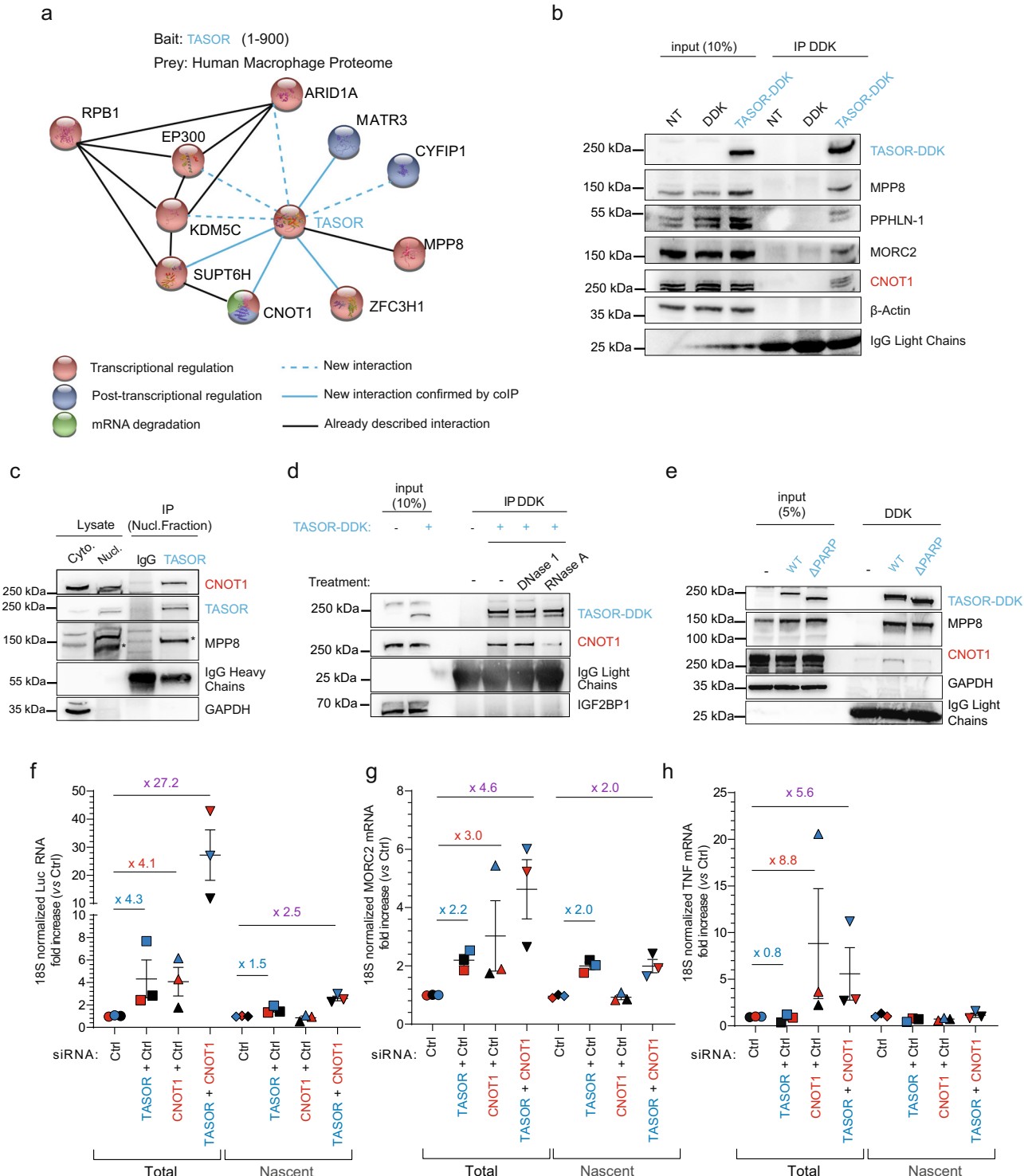

involved in epigenetic repression of retrotransposons and RNA-dependent suppression of intergenic transcription[24,25] (Fig. S2a, b).

**TASOR interacts and cooperates with the CCR4-NOT complex scaffold CNOT1**. To obtain insights into the pathways contributing to TASOR post-transcriptional functions, we performed a yeast two-hybrid (Y2H) screen using the first 900 aa of the 1670 aa-long TASOR protein as bait, and the proteome from human macrophages as prey, since this cell type represents a natural target for HIV. As expected, we found that the TASOR

N-terminal region binds its known HUSH partner, MPP8. Other candidate binding-proteins are factors involved in gene transcription modulation, LXRα, ARID1A (also known as BAF250), EP300, SUPT6H, KDM5C, as well as proteins involved in mRNA metabolism, CYFIP1, CNOT1, DHX29, DHX30, EIF4G1, ZFC3H1, and MATR3 (Fig. S2c). The latter has already been discovered as a TASOR interacting protein in previous large-scale interactome studies[26,27]. Among these candidate partners of TASOR, some are negative regulatory factors of gene expression directly connected to RNA pol II (RNAPII) subunit RPB1 (Fig. 2a), which may indicate a role of TASOR and partners in

**Fig. 2 TASOR interacts and cooperates with the CCR4-NOT complex scaffold CNOT1 to destabilize LTR-driven transcripts. a** Identification of CNOT1 and other proteins as new partners of TASOR by yeast-two-hybrid screening. Proteins involved in transcriptional regulation, in post-transcriptional regulation of RNAs, and in mRNA degradation pathways are colored in red, blue, and green, respectively. Newly found interactions are shown in blue. Those validated by at least 3 coIPs are shown in solid line (TASOR-CNOT1 validated in Figs. 2b–e, 3a, 6b–d, 7g, and S3c, TASOR-SUPTH6 in Fig. S2g, TASOR-MATRIN 3 in Fig. 6d, TASOR-ZFC3H1 in Fig. S3c). **b** TASOR interacts with CNOT1 by co-immunoprecipitation. Mock, DDK, and TASOR-DDK vectors were transfected in HeLa cells and anti-DDK immunoprecipitation was performed ($n > 3$, some are presented all along this study). **c** Endogenous TASOR interacts with endogenous CNOT1 in the nucleus of HeLa HIV-1 LTR-ΔTAR-Luc cells. GAPDH is a negative control. ($n = 2$); **d** a transcript stabilizes TASOR-CNOT1 interaction. DDK and TASOR-DDK vectors were transfected in HeLa cells. Lysates were treated or not with DNase 1 or RNase A. Anti-DDK immunoprecipitation was then performed. IGF2BP1 is a negative control ($n > 3$). **e** The PARP domain of TASOR is required for its interaction with CNOT1. HeLa HIV-1 LTR-ΔTAR-Luc cells were transfected with pLenti-TASOR WT-DDK or TASORΔPARP-DDK vectors. An anti-DDK IP and a western blot analysis were performed to assess the interactions with CNOT1 ($n > 3$). **f** TASOR and CNOT1 cooperate to repress HIV-1 LTR expression at a post-transcriptional level. After siRNA transfection in HeLa HIV-1 LTR-ΔTAR-Luc, a *Nuclear Run On* experiment was undertaken to measure LTR-driven Luc transcripts levels. ($n = 3$; each color represents one different independent experiment, mean and SEM are shown). **g** TASOR represses MORC2 transcription while CNOT1 decreases its stability. After siRNA transfection in HeLa HIV-1 LTR-ΔTAR-Luc, a *Nuclear Run On* experiment was undertaken to measure MORC2 transcripts levels ($n = 3$; each color represents one different independent experiment, mean and SEM are shown). **h** CNOT1 silencing only increases TNFα transcript stability. After siRNA transfection in HeLa HIV-1 LTR-ΔTAR-Luc, a *Nuclear Run On* experiment was undertaken to measure TNFα transcripts levels ($n = 3$; each color represents one different independent experiment, mean and SEM are shown). Source data are provided as a Source data file.

inhibiting gene expression during the transcription process per se. We chose to focus on CNOT1, which is the scaffold protein of the most important and conserved deadenylase complex from Yeast to Human, the Carbon catabolite repression 4-negative on TATA-less complex or CCR4-NOT[28,29]. Interestingly, CNOT1 was found in a siRNA screen for cellular factors involved in MMLV-*Gfp* reporter silencing in ESCs, along with MPP8 or ATF7IP, a protein essential for heterochromatin formation by HUSH[30,31]. We confirmed the interaction between epitope-tagged TASOR (TASOR-DDK) and endogenous CNOT1, in addition to already known TASOR-binding proteins such as MPP8, Periphilin, and MORC2 (Fig. 2b), and between endogenous TASOR and endogenous CNOT1 (Figs. 2c and S2d). While DNase I-treatment does not prevent TASOR and CNOT1 interaction, RNase A-mediated RNA digestion reduces the amounts of co-immunoprecipitated CNOT1 in HeLa cells (Fig. 2d), suggesting that RNA favors the interaction between the two partners. Importantly, the inactive TASORΔPARP mutant does not interact with CNOT1, which is consistent with the hypothesis that TASOR-CNOT1 association is involved in TASOR-mediated repression (Fig. 2e). CNOT1 being a scaffold protein, we checked whether TASOR could interact with other components of the CCR4-NOT complex. By immunoprecipitating DDK-tagged CNOT7, the deadenylase partner of CNOT1, we revealed an interaction with endogenous TASOR, along with the other CCR4-NOT complex CNOT9 and CNOT1 components (Fig. S2e).

Since TASOR and CNOT1 interact with each other, we tested whether they collaborate to repress LTR-driven expression at the level of the transcription per se and/or at the level of RNA stability by performing *Nuclear Run On* experiments as in Fig. 1. Silencing of TASOR and/or CNOT1 was checked by western blot (Fig. S2f). The main effect of CNOT1 was detected at the total level of LTR-driven transcripts, whose RNA amounts were increased by 4-fold upon CNOT1 silencing, in agreement with its known role on RNA stability (Fig. 2f). Interestingly, when silencing both TASOR and CNOT1, a strong synergistic effect was observed at the total RNA level, suggesting that the two proteins act in different pathways but cooperate to repress the expression of the transcript particularly at a post-transcriptional level (Fig. 2f: 27.2- vs 2.5-fold increase on total vs nascent RNA). As discussed above, siRNA or Vpx-mediated TASOR depletion only increased the level of *MORC2* mRNAs at the transcriptional stage (no difference between total and nascent RNA—Figs. 2g and S1c). In contrast, CNOT1 depletion did not increase nascent

*MORC2* mRNA levels but led to its accumulation (Fig. 2g, 0.9- vs 3.0-fold, respectively). On the other hand, CNOT1 has been shown to interact with *TNFα* mRNA and to induce its deadenylation with CNOT7[32–35]. Accordingly, we confirmed that CNOT1 silencing increased by 8.8-fold the total *TNFα* mRNA level without influencing *TNFα* transcription (Fig. 2h). Of note, combining siCNOT7 with siCNOT1 does not further increase Luc expression as compared to siCNOT1 alone, in agreement with CNOT1 and CNOT7 belonging to the same pathway (Fig. S2g). However, the inhibition of both CNOT7 and TASOR expression suggests a collaboration between CCR4-CNOT and TASOR (Fig. S2g). Altogether, our results show that TASOR and CCR4-NOT interact both physically and functionally and can strongly cooperate at a post-transcriptional level to decrease LTR-driven transcript accumulation.

**TASOR-CNOT1 complex represses LTR-driven expression in two models of HIV-1 latency**. Next, we questioned the relevance of the TASOR-CNOT1 cooperation in more relevant HIV models. First, we confirmed that endogenous TASOR and CNOT1 interact in a Jurkat-derived latency model for HIV-1 (J-Lat A1, Fig. 3a). This T-cell line contains an HIV-1 LTR-*tat-IRES-gfp*-LTR minigenome stably integrated at a unique site and epigenetically silenced[36]. These cells were transduced with a vector expressing a doxycycline-inducible miRNA directed against CNOT1 mRNA or, as control, a miRNA against luciferase mRNA (Fig. 3b). The addition of doxycycline reactivates green fluorescent protein (GFP) expression in miR-CNOT1 cells, suggesting that CNOT1 contributes to HIV-1 silencing in J-Lat A1 cells (Fig. 3c). Then, TASOR and CNOT1 were both silenced by delivering Vpx and by adding doxycycline, respectively, to cells treated with TNFα, which exacerbates the effect of TASOR depletion[5]. The Vpx R42A mutant, unable to trigger TASOR degradation, was used as a control. Western blot analysis confirmed the depletion of TASOR induced by WT Vpx, but not by Vpx R42A, and the partial downregulation of CNOT1 in the presence of doxycycline only in the miR-CNOT1 cells (Fig. 3d). The knock-down of both TASOR (by Vpx) and CNOT1 (upon doxycycline induction) appears to increase GFP expression to higher extent than the downregulation of each factor separately (Fig. 3e). Indeed, quantification of *GFP* mRNA levels indicates a synergistic effect of TASOR and CNOT1 on the reactivation of the GFP reporter (Fig. 3f, 20.2× fold increase when silencing both TASOR and CNOT1 expressions in comparison to 4.3× and 2.9× upon TASOR and CNOT1 silencing, respectively). In contrast,

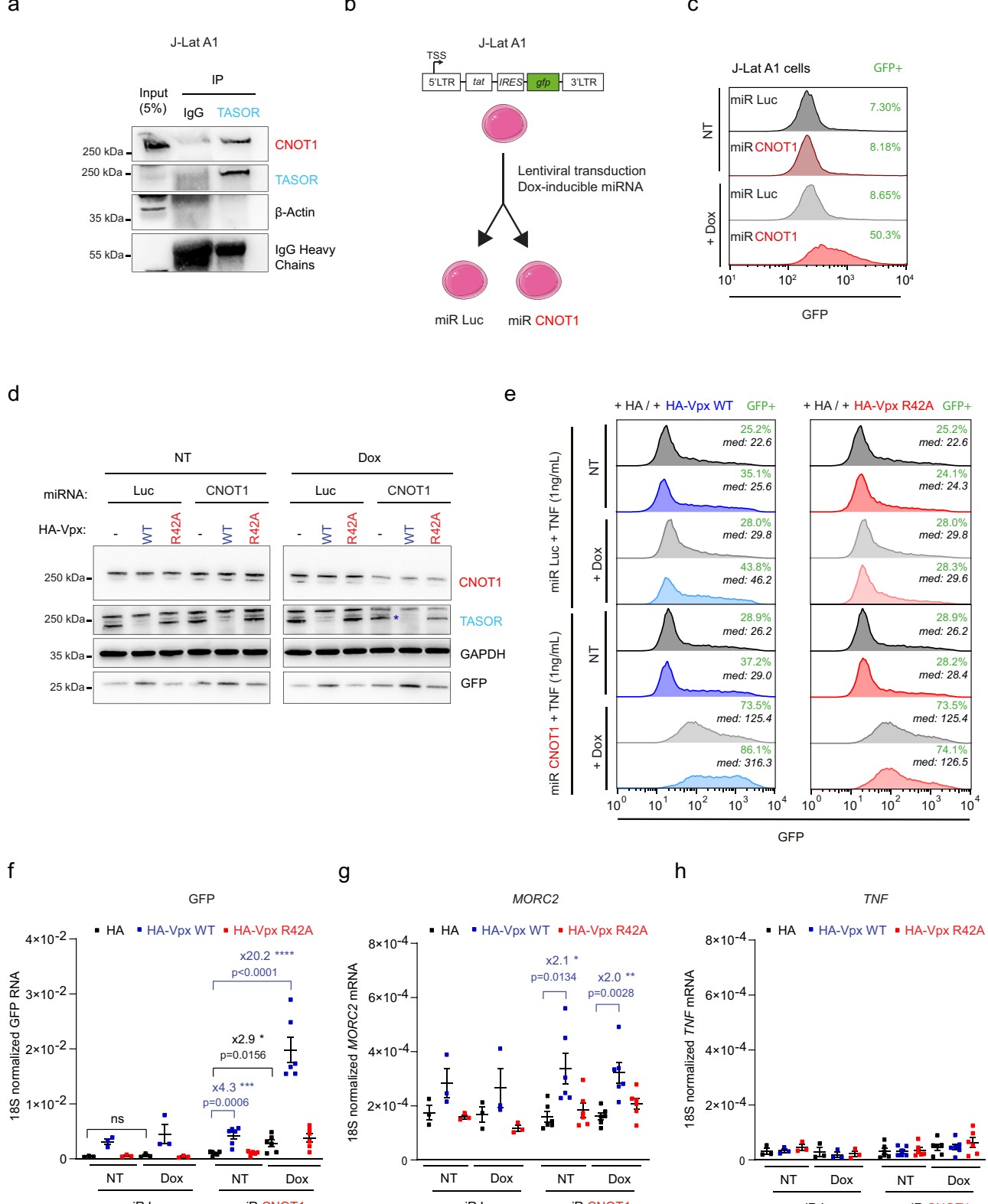

*MORC2* mRNA expression levels are solely dependent on TASOR expression under these experimental conditions, whereas *TNFα* mRNAs remain very lowly expressed under all conditions (Fig. 3g, h).

Second, we tested the TASOR-CNOT1 synergy under conditions where infected cells represent a population with diverse integration sites as previously achieved[13]. Jurkat T cells were

infected with a genetically marked HIV-1 virus[13] (Fig. 4a). This vector retains the complete LTRs, *tat,* and *rev*, but has a frameshift mutation in *env, ngfr* in place of *nef* and *egfp* in place of *gag, pol, vif*, and *vpr*. Finally, the splicing sites are conserved. We sorted infected cells by cytometry (EGFP-positive cells), kept the sorted cells in culture for another two weeks until EGFP expression was silenced, and finally sorted the population of

**Fig. 3 TASOR and CNOT1 cooperate to repress HIV-1 expression in the J-Lat A1 model of HIV-1 latency. a** TASOR interacts with CNOT1 in the J-Lat A1 T cell line. TASOR or Rabbit IgG immunoprecipitations were performed in J-Lat A1 lysates. The interaction was then assessed by western blot. β-Actin is a negative control. (*n* = 2 in Jurkat cell lines). **b** Generation of Dox-inducible miR-RNA CNOT1 J-Lat A1 cells. J-Lat A1 cells, harboring one copy of integrated and latent LTR-tat-*IRES*-GFP-LTR construct, were transduced with a pINDUCER10 vector containing a doxycycline-inducible miRNA targeting CNOT1 or Luciferase (Mock) transcripts. **c** Expression of the miRNA targeting CNOT1 reactivates GFP expression from the latent HIV-1 minigenome. J-Lat A1 miR Luc or miR CNOT1 were treated or not with doxycyline 1 µg/mL for 72 h and GFP expression was analyzed by flow cytometry. The proportion of cells that became GFP positive is indicated in green. **d** Dox-inducible miR-CNOT1, but not miR-Luc, decreases CNOT1 expression while delivery of VLPs containing HIV-2 Vpx WT induces TASOR depletion in contrast to Vpx R42A in the J-Lat A1 cells. J-Lat A1 miR Luc and miR CNOT1 were treated or not for 72 h with doxycycline 1 µg/mL and VLPs containing HA-Vpx WT or R42A or mock were delivered for 24 h prior to protein analysis. **e–f** Co-silencing of CNOT1 and TASOR synergistically increases LTR-driven GFP expression at the protein level with the proportion of cells that became GFP positive indicated in green and the median of GFP expression in the whole cell population in italic (**e**), at the RNA level (**f**), but not MORC2 RNA (**g**) or TNF RNA (**h**) in the J-Lat A1 model. J-Lat A1 miR Luc and miR CNOT1 were treated or not for 72 h with doxycycline 1 µg/mL and VLPs containing HA-Vpx WT or R42A or mock were delivered for 24 h and TNF (1 ng/mL) 16 h prior to flow cytometric analysis and RT-qPCR analyses (from **d** to **h**, *n* = 3 for miR Luc; *n* = 6 for miR CNOT1; each independent replicate, mean and SEM are shown; two-sided unpaired *t* test was applied). Source data are provided as a Source data file.

latently infected cells (EGFP-negative cells; Fig. 4b, left). These cells do not express EGFP at the protein level whereas HIV-1-unspliced RNA could be detected by reverse transcriptase quantitative polymerase chain reaction (RT-qPCR), suggesting that a co- or post-transcriptional repressive mechanism could contribute to the latent state (Fig. 4b, right). To assess the potential cooperation between TASOR and CNOT1 in this model of HIV-1 latency, we depleted TASOR and CNOT1 by Vpx and miR-CNOT1, respectively, as previously (Fig. 4c). Analysis of EGFP expression by cytometry or quantification of unspliced RNA transcribed from the HIV-1 5′LTR demonstrate the synergy between TASOR and CNOT1 in repressing HIV-1 LTR-driven expression in T cells (Fig. 4d, e, 16.1× fold increase when silencing both TASOR and CNOT1 expressions in comparison to 3.0× and 3.6× upon TASOR and CNOT1 silencing respectively). Only a twofold increase in *MORC2* transcript levels (corresponding to TASOR silencing) and no effect on the *TNFα* transcripts were observed under these experimental conditions (Fig. 4f, g).

**A set of host genes are cooperatively regulated by TASOR and CNOT1.** To assess whether TASOR and CNOT1 can cooperate beyond HIV repression, we conducted polyA+ RNA-seq in HeLa cells silenced for TASOR, CNOT1, or both (Fig. 5a), and we investigated the impact of these alterations on the expression of host genes and transposable elements, including LINE-1 (L1).

Three hundred and ninety-five host genes were upregulated both by the dual depletion of TASOR and CNOT1 and by each individual knock-downs (Fig. 5b). Among them, we selected those with a significant expression level (transcripts per million (TPM) > 1), and we modeled the interaction factor between siTASOR and siCNOT1 conditions to quantitatively assess potential cooperation between these two factors (Fig. 5c). Synergistic, additive, and negative interactions reflect situations where the expression fold change obtained by the double depletion is higher, equal, or lower, respectively, than the mere product of the fold changes obtained by individual RNA depletion. More than 200 genes are significantly and cooperatively regulated by TASOR and CNOT1 (64 by synergistic effects, 168 by additive effects, Fig. 5d). Then we looked at the expression of repetitive sequences at the family level. In HeLa cells, and in the short-term after knock-down, the effect of TASOR depletion, alone or in combination with CNOT1 depletion, does not lead to upregulation of L1 elements and has a limited effect on other repeats (Fig. 5e)[7–10,37]. Only three repeat families are additively upregulated by TASOR and CNOT1 co-silencing: LTR1, LTR12C, and HERV9NC-int (Fig.5f). Of note LTR12C and HERV9NC-int represent the LTR and internal sequence of the same element, a variant of the HERV-9 family, respectively and ERV9-LTR12 was already shown to be the most derepressed LTR

upon MPP8-inactivation[37]. Altogether, these results suggest that the TASOR-CNOT1 cooperation can operate not only on HIV LTR but also on host genes.

**TASOR interacts and cooperates with nuclear RNA degradation factors.** To gain insight into the mechanism of TASOR-CNOT1-mediated repression mechanism, we questioned the involvement of MTR4 present in the human TRAMP, NEXT or PAXT complexes, but also human EXOSC10 (RRP6 in yeast) from the exosome and the m⁶A reader YTHDF2. Indeed, in yeast, Not1, the homolog of human CNOT1, has been shown to recruit Mtr4 and the exosome, resulting in the degradation of nuclear defective and nascent RNAs[38,39]. In addition, MTR4 and EXOSC10/RRP6 are members of an RNA surveillance complex that inhibits LTR-driven expression in the HeLa HIV-1 WT LTR-Luc model[40]. Finally, YTHDF2 was reported on the one hand to interact with CNOT1, which leads to the destabilization of m⁶A-modified mRNAs[41] and on the other hand to be recruited onto HIV-1 5′LTR, Nef, and 3′LTR genomic RNA sequences[42,43]. By cell fractionation, we show that TASOR protein is predominantly detected in the nucleus, while CNOT1 is a shuttling protein, present both in the cytoplasm and nucleus, in agreement with previous studies[44–46] (Fig. 6a). As controls, the MATR3 and GAPDH proteins fractionated in the nucleus and the cytoplasm, respectively, as expected (Fig. 6a). Then, in an endogenous TASOR immunoprecipitate from nuclear extracts, we could retrieve EXOSC10, MTR4, and YTHDF2, in addition to CNOT1 and CNOT7 (Fig. 6b) and, in a CNOT1 immunoprecipitate, we could retrieve CNOT7 and YTHDF2 as expected, as well as MTR4, EXOSC10 and TASOR (Fig. 6c). The same components were pulled-down in an MTR4 immunoprecipitate except for CNOT7 (Fig. 6c). Next, we overexpressed TASOR-DDK in cells and performed a DDK immunoprecipitation to recover TASOR and its bound partners in the absence or in presence of RNase A (Figs. 6d and S3a). MATR3 is the only protein along with MPP8, our positive control, to be efficiently co-immunoprecipitated with TASOR under RNase A treatment, which confirms our Y2H results and that the interactions between TASOR and MATR3, MPP8 are RNA independent (Fig. 6d). However, the interactions between TASOR-DDK, CNOT1, and its known partners MTR4, YTHDF2 seem to rely on the presence of RNAs since TASOR affinity for these proteins is decreased upon RNase A treatment. (Fig. 6d).

To verify the involvement of these new factors in the TASOR repression of LTR expression, we silenced these proteins expressions alone or in combination with TASOR and measured Luciferase activity (Fig. 6e). The strongest synergistic effect was always obtained under TASOR and CNOT1 co-silencing (Fig. 6e). Nonetheless, we observed that TASOR also cooperates with

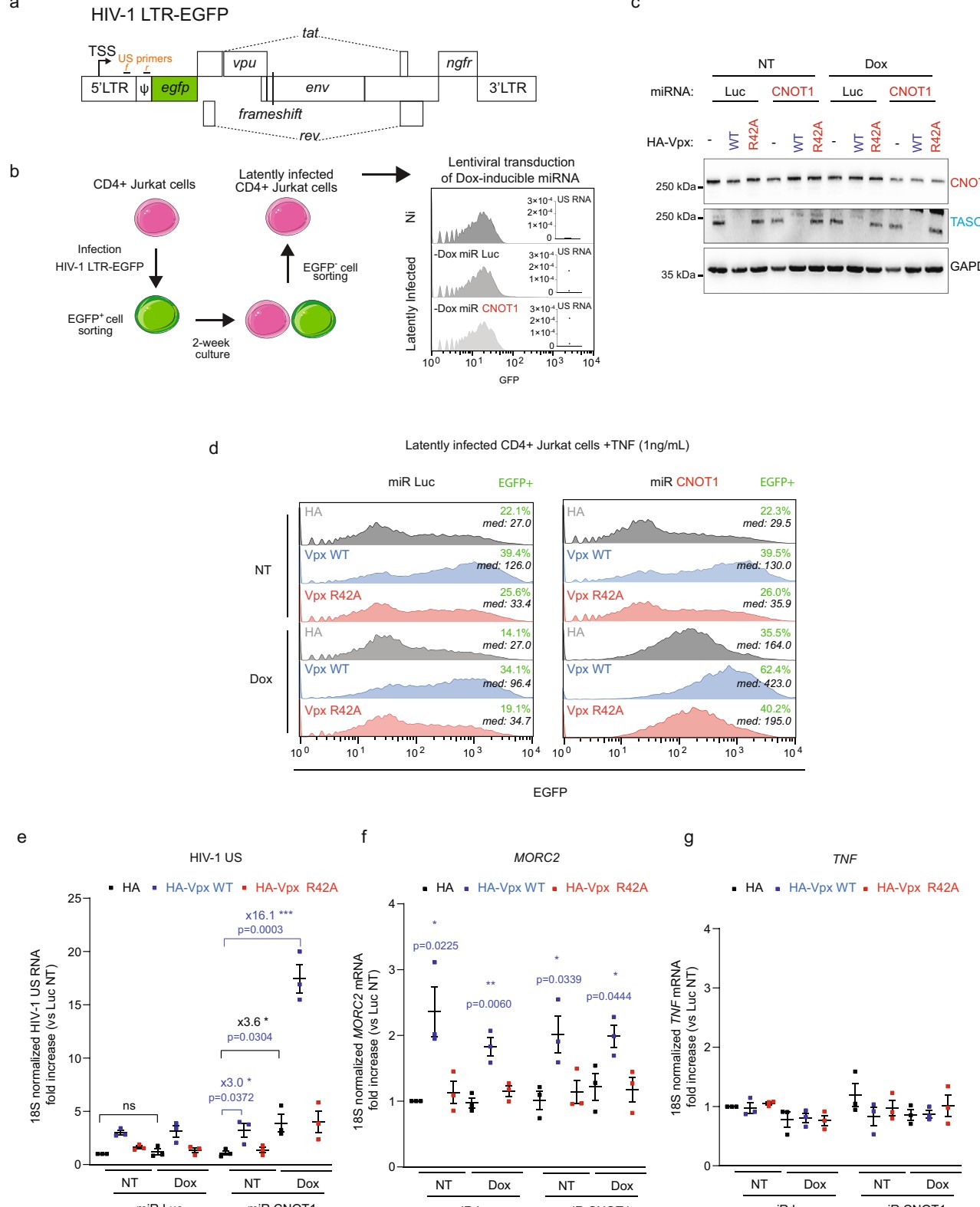

MATR3, YTHDF2, MTR4 (Fig. 6e), or with EXOSC10 (Fig. S3b). In humans, MTR4 is found in at least three different complexes that use the exosome complex to eventually degrade the targeted RNA: the nucleolar TRAMP complex, the nucleoplasmic TRAMP-like NEXT and PAXT complexes[47] (reviewed in ref. [48]). Of note, in our Y2H screen, we also identified ZFC3H1, a member of the PAXT complex, as a partner of TASOR (Fig. S2c). We confirmed, in nuclear immunoprecipitates, that TASOR interacts with the two main members of the TRAMP-like PAXT complex MTR4 and ZFC3H1 (Fig. S3c). Interestingly, ZFC3H1 has already been discovered as one of Periphilin's partners[47,49]. In conclusion, our results suggest that TASOR and CNOT1 together form a platform recruiting different factors involved in RNA degradation pathways (Fig. 6f).

**Fig. 4 TASOR and CNOT1 cooperate to repress HIV-1 expression in latently HIV-1-infected Jurkat T cells. a** Schematic of the LTR-EGFP provirus used to analyze HIV-1 LTR-driven EGFP expression in infected Jurkat T cells. Primers used to quantify LTR-driven transcripts are mapped in orange. **b** Generation of latently HIV-1-infected Jurkat cells based on the EGFP expression by flow cytometry. The sorted cells were then transduced with the doxycycline-inducible miR CNOT1 or Luc vectors. Without doxycycline treatment, cells remained EGFP negative by flow cytometry, whereas LTR-driven RNAs (US) were detected by RT-qPCR in the latently infected cells. RNA was normalized on the 18s rRNA ($n = 2$). **c** Dox-inducible miR-CNOT1, but not miR-Luc, decreases CNOT1 expression while the delivery of VLPs containing HIV-2 Vpx WT induces TASOR depletion in contrast to Vpx R42A in these latently infected Jurkat cells. The latently HIV-1-infected Jurkat miR Luc and miR CNOT1 cells were treated or not for 72 h with doxycycline 1 μg/mL, with VLPs containing HA-Vpx WT or R42A or mock were delivered for 24 h and with TNF (1 ng/mL) 16 h prior to protein analysis. The co-depletion of CNOT1 and TASOR synergistically increases HIV-1 LTR at the protein level with the proportion of cells that became EGFP positive indicated in green and the median of GFP expression in the whole cell population in italic (**d**), at the RNA level (**e**), but not MORC2 RNA (**f**) or TNF RNA (**g**) in these latently HIV-1-infected T cell lines. The latently HIV-1-infected Jurkat miR Luc and miR CNOT1 cells were treated or not for 72 h with doxycycline 1 μg/mL and VLPs containing HA-Vpx WT or R42A or mock were delivered for 24 h and TNF (1 ng/mL) 16 h prior to flow cytometric analysis and RT-qPCR analyses (for **d**–**g**, $n = 3$ independent experiments, mean and SEM are represented; two-sided unpaired $t$ test was applied: $p$ values are indicated in the graph). Source data are provided as a Source data file.

**TASOR recruits RNA-degradation factors onto elongating RNAPII.** The ability of TASOR to propagate H3K9me3 marks and its association with CNOT1, with subunits of a TRAMP-like/PAXT complex, and with the exosome led us to envision that HUSH might function similarly to the fission yeast RNA-induced silencing complex (RITS). Indeed, RITS induces transcriptional silencing by adding H3K9me3 marks concomitantly to the synthesis of a transcript and its processing by a TRAMP complex and the exosome[50]. Alternatively, in fission yeast, silencing can depend on recognition of the transcript by small interfering RNAs produced by Dicer dsRNA-cleavage activity[51,52]. The inhibition of DICER by siRNAs does not increase LTR-driven Luc expression in our experimental model (Fig. S3d) suggesting that the Dicer-dependent RNAi pathway is not involved here. In addition, RITS is constituted of Chp1 and Tas3, with Chp1 harboring a SPOC domain alike TASOR and a trimethyl-binding domain, the chromodomain, alike MPP8[53,54]. Therefore, we hypothesized that HUSH could propagate epigenetic marks by following RNA polymerase II. We further tested this possibility by assessing the presence of TASOR at transcriptional centers by single-molecule RNA fluorescence in situ hybridization (smFISH) in the HIV-1 LTR-EGFP model of latency used in Fig. 4. In these experiments, cells were co-treated with TNF and Virus-Like Particles (VLPs) containing Vpx WT or Vpx R42A (Fig. S4a, b) to confirm the specificity of TASOR staining, while we labeled HIV-1 transcripts with probes against *egfp* (Fig. S4c). In the presence of Vpx R42A, TASOR is enriched at transcriptional centers, characterized by large viral RNA spots, but not in all cells, as expected from position-effect variegation (Fig. 7a, left, and Fig. 7b). However, in the presence of WT Vpx, the TASOR signal was significantly reduced at transcriptional centers (Fig. 7a, right, and Fig. 7b) while these Jurkat cells show a 2.7-fold increase in HIV-1 transcriptional centers compared to cells treated with the degradation-defective Vpx mutant (Fig. S4d). Vpx-mediated degradation of TASOR also promotes a 2.6-fold increase of the number of viral RNA molecules in the nucleus (Fig. S4d) and an increase in EGFP reporter synthesis (Fig. S4e). In addition, we could co-immunoprecipitate RNAPII with TASOR (Fig. 7c). In contrast, TASORΔPARP very weakly interacts with RNAPII, in correlation with its inability to repress LTR-driven expression (Fig. 7d). More importantly, we uncover that TASOR interacts predominantly with a phosphorylated form of RNAPII, PhosphoSer2-RNAPII, specific for the elongation phase of transcription, rather than with the phosphorylated form enriched during transcription initiation, PhosphoSer5-RNAPII (Fig. 7e). The analysis of CUT&Tag and CUT&RUN data from Douse et al.[9] combined with ChIP-seq and RNA-seq data from Liu et al.[55] supports our hypothesis, as we could find several examples of genes covered by both RNAPII and TASOR-dependent H3K9me3 marks (Fig. S4f, g). Next, we used formaldehyde-

assisted isolation of regulatory elements coupled with qPCR (FAIRE-qPCR), which allows distinguishing nucleosome-depleted DNA regions from the bulk of chromatin. While TASOR silencing had no impact on the so-called Nuc0 nucleosome positioned at the very beginning of the LTR promoter, it increases chromatin accessibility of the LTR-Nuc1 region downstream of the transcription start site and triggers a dramatic decompaction of the Luc coding sequence, even more pronounced than the change induced by TNFα itself (Fig. S4h, i). Strikingly, overexpression of TASOR enhances the affinity of CNOT1, MTR4, EXOSC10, YTHDF2, and MORC2 to RNAPII but not that of the RNAPII elongation cofactor SUPT6H (Fig. 7f). Furthermore, after BrUTP labeling of nascent transcripts, we coupled immunoprecipitation of endogenous TASOR or CNOT1 to RT-qPCR and found that the nascent LTR-derived Luc transcript was equivalently associated with TASOR and CNOT1, which underlines their direct functions on HIV-1 LTR-derived transcript, whereas the nascent *MORC2* transcript and Taurine-upregulated gene 1 lncRNA, which we identified as TASOR target in the data from Liu et al.[10] and Douse et al.[9], were mainly associated with TASOR (Fig. 7g). Altogether, these results suggest that TASOR epigenetic repressor helps assemble RNA metabolism machineries on RNAPII during transcription elongation and targets the nascent RNA.

**Discussion**

Overall, our results support a model in which TASOR, in association with CNOT1, provides a platform along transcription to destabilize nascent transcripts. This conclusion is supported by: (i) the effect of TASOR beyond transcription per se at a post-transcriptional level; (ii) the interaction of TASOR with CNOT1 and their synergistic repressive effect on HIV-1 LTR-driven expression; (iii) the interaction of TASOR with members of the TRAMP-like/PAXT complex, ZFC3H1 and MTR4, and with the RNA exosome component EXOSC10, as well as its functional cooperation with them; (iv) the fact that the interactions between TASOR and RNA degradation factors are stabilized in the presence of RNA; (v) the decompaction of the coding sequence under TASOR silencing; (vi) the interaction of TASOR with elongating RNAPII; (vii) the recruitment of RNA degradation factors to RNAPII when TASOR is overexpressed, and (viii) the accumulation of TASOR in transcription centers and the interaction of TASOR and CNOT1 with nascent transcripts. While indirect effects cannot totally be ruled out from siRNA, miRNA or Vpx-mediated deletion experiments, this latter point suggests a direct effect of TASOR and CNOT1 on LTR-derived RNA metabolism.

Consistent with our observations, RPRD2, a regulator of RNAPII, together with RNA metabolism proteins were also identified as TASOR partners in a BioID assay[9]. Keeping in mind

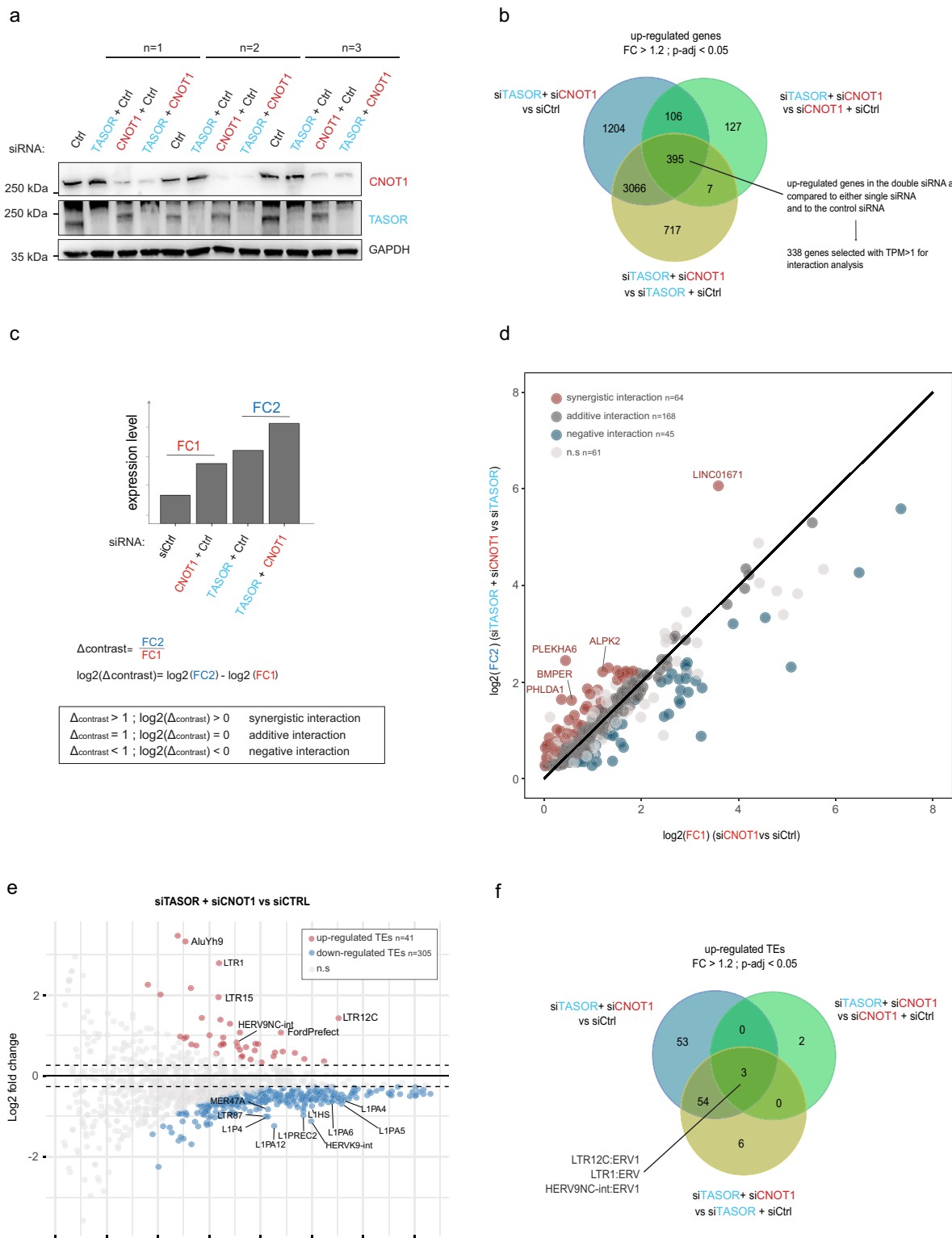

the role of TASOR at the epigenetic level[6], we propose a feedback control mechanism in which TASOR and CNOT1 would follow RNAPII during HIV-1 provirus transcription by association with the nascent transcript and recruit RNA metabolism proteins, leading in turn to the degradation of the transcript and the deposition of H3K9me3 marks (Fig. S5b).

This molecular defense mechanism was not already described in mammals although it seems to be quite well conserved throughout the evolution of eukaryotes in unicellular organisms such as *Schizosaccharomyces pombe* or more complex organisms such as *Drosophila melanogaster* or *A. thaliana* to preserve genome integrity. Indeed, our model parallels the gene expression repression mechanism proposed for the RITS complex in fission yeast or for the piRNA-guided transcriptional silencing Piwi-Asterix-Panoramix-Eggless complex model in drosophila[56–58]. In each case, transcriptional epigenetic repression is coupled to the

**Fig. 5 TASOR cooperates with CNOT1 to repress a subset of genes in HeLa cells. a** HeLa cells were transfected for 72 h with siRNA targeting TASOR or CNOT1 or both transcripts. A western blot assessed the downregulation of TASOR and CNOT1 prior to RNAseq analyses. (western blot of the three different replicates is presented). **b** Number of genes that are significantly upregulated (by at least 20%), upon TASOR + CNOT1 co-silencing as compared to control, TASOR, or CNOT1 individual silencing in HeLa cells (log2 fold change > 0.2630344; *p* adjusted values < 0.05 after Benjamini–Hochberg multiple testing correction of Wald test *p* value). **c** Cooperativity of TASOR and CNOT1 regulation on host genes. The interaction term of the model used for differential gene expression analysis (Δcontrast, DESeq2). **d** Cooperative effects affecting genes upregulated in one or the other of the single siRNA conditions (siTASOR or siCNOT1) and not downregulated in the second condition (log2 fold change > 0; *p* adjusted values < 0.05 after Benjamini–Hochberg multiple testing correction of Wald test *p* value). Fold changes (FC1 and FC2, as defined above) for individual genes were plotted showing the potential interaction modes of TASOR and CNOT1. **e** MA plot representing the differential expression of transposable element (TE) families in HeLa cells, upon co-silencing of TASOR and CNOT1 as compared to cells transfected with control siRNA. Significantly upregulated and downregulated TE families are highlighted in red and blue, respectively (log2(FC) > 0.2630344; *p* adjusted values < 0.05, Wald test with Benjamini–Hochberg multiple testing correction). ns not significant. **f** Number of TE families that are significantly upregulated (by at least 20%), upon TASOR + CNOT1 co-silencing as compared to control, TASOR, or CNOT1 individual silencing in HeLa cells (log2 fold change > 0.2630344; *p* adjusted values < 0.05 after Benjamini–Hochberg multiple testing correction of Wald test *p* value). More information is available in Source data file.

synthesis of a transcript and sometimes to its degradation. The fission yeast RITS complex seems to match well our HUSH model, with yeast Chp1 and Tas3 sharing structural features with TASOR, MPP8 and PPHLN-1, as also noticed by Douse et al[9,53] and with the ability of RITS to interact with Not1/CNOT1[59], the exosome and MTR4 alike HUSH (Fig. S5a, b). In addition, Skalska et al. recently showed that HUSH is recruited to chromatin upon RNAPII inhibition or RNase treatment, suggesting that RNA degradation of the nascent transcript could modify HUSH binding to chromatin, perhaps due to binding competition between nascent RNA and methylated histones, in agreement with our model[60]. Then, the role of HUSH in transcript destabilization could also be disconnected from its ability to epigenetically repress HUSH target genes. Indeed, in pluripotent stem cells, MPP8 lacking its chromodomain, which permits its binding to H3K9me3 marked chromatin, can still repress LINE-1 elements[61], suggesting that in this configuration, HUSH could repress its targets independently from chromatin binding and maintenance of H3K9me3. It is not yet known whether the reverse, i.e. repression at the epigenetic level unrelated to RNA degradation mechanisms, is also possible. Therefore, a fundamental unanswered question is whether HUSH-mediated epigenetic repression could be effective by itself or whether HUSH must always rely on RNA degradation factors recruitment to enforce HIV silencing.

Inhibition of DICER has no effect on LTR-driven expression in our experimental system, suggesting that DICER-produced small interfering RNAs are not involved in HUSH-mediated repression of HIV-1, while small RNAs are known to mediate chromatin silencing in fission yeast or *D. melanogaster* for instance[62]. However, we cannot rule out the possible involvement of small interfering RNAs in the silencing of other HUSH gene targets.

To extend our observations beyond HIV transcription, we performed a transcriptomic analysis of cells silenced for TASOR, CNOT1, or both. Our data highlight that TASOR and CNOT1 cooperate to regulate a number of cellular genes, suggesting that a RITS-like mechanism could also apply to endogenous cellular targets. To our surprise, we do not detect upregulation of L1 RNAs following TASOR silencing, as expected from previous observations in other cell types[7–10,37], suggesting that additional and predominant repressive mechanisms may operate in HeLa cells to silence L1 elements, such as DNA methylation[63]. Alternatively, long-term shRNA silencing or full knock-out by CRISPR may be required to fully reveal the entire landscape of HUSH-regulated transcripts[10,37].

Regarding HIV, our results with two models of HIV-1 latency may suggest HUSH to be involved in the maintenance of HIV latency. Nonetheless, the post-transcriptional activity of HUSH

along with RNA synthesis suggests that HUSH activity needs the production of the viral RNA. However, we were unable to reveal a repressive role for HUSH in biological systems of high sustained viral RNA transcription, either due to proviral integration into transcriptionally active chromatin sites or due to high levels of the Tat protein. Since HUSH repression is dependent on the integration site[6], it is conceivable that HUSH triggers H3K9 tri-methylation—or its maintenance—only on provirus sequences integrated into poorly, but still, transcribing regions with signals of heterochromatinization. In this spatial window HUSH could counteract stochastic bursts of expression from the lowly expressed viral promoter[64].

Our study also opens questions concerning the role of HIV-2 Vpx. By inducing HUSH degradation, Vpx is able to increase the proviral transcription from the HIV LTR and the stability of the LTR-driven transcript, thus Vpx hinders the silencing of the provirus. One could reasonably think at first that Vpx confers an advantage in HIV-2 replication. Though HIV-2 is able to counteract several restriction factors, such as SAMHD1, APOBEC3G and BST2, HIV-2 replication is still very weak as compared to HIV-1 that does not counteract SAMHD1 or HUSH. As a repressor of retroelements expression, HUSH serves as a guardian of the integrity of the host genome. Then, by counteracting HUSH to increase HIV proviral expression, Vpx could, at the same time and as collateral damage, induce genomic deregulation and establish an unfavorable environment that subsequently limits HIV-2 replication. Nonetheless, directly linking clinical presentations to restriction factors activity may overlook other biological processes, including the immune response triggered by other components of the virus and the replication cycle itself. Overall, our study provides new insights into how HUSH represses retroelements such as the HIV provirus. We propose that TASOR acts as a hub linking the elongating RNAPII, the epigenetic repressor HUSH, and RNA degradation factors to repress the transgene both at the transcriptional level by the deposition of H3K9me3 marks and at the post-transcriptional level to degrade the synthesized target RNA. As a result, the host invader remains poorly expressed. Excitingly, the study by Jiang et al.[65], shows that, in elite controllers, intact HIV-1 proviruses are present in H3K9me3-rich heterochromatin such as centromeric regions and in KRAB-Zinc finger genes, which are targeted by HUSH[6]. Understanding how HUSH is precisely recruited to provirus sequences or retroviral RNA could help decipher its role in the establishment of latent HIV reservoirs in infected patients.

## Methods
**Plasmids**. TASOR expression vectors pLenti-myc-DDK, pLenti-TASOR-myc-DDK were purchased from Origene. pLenti-TASOR-myc-DDK expresses

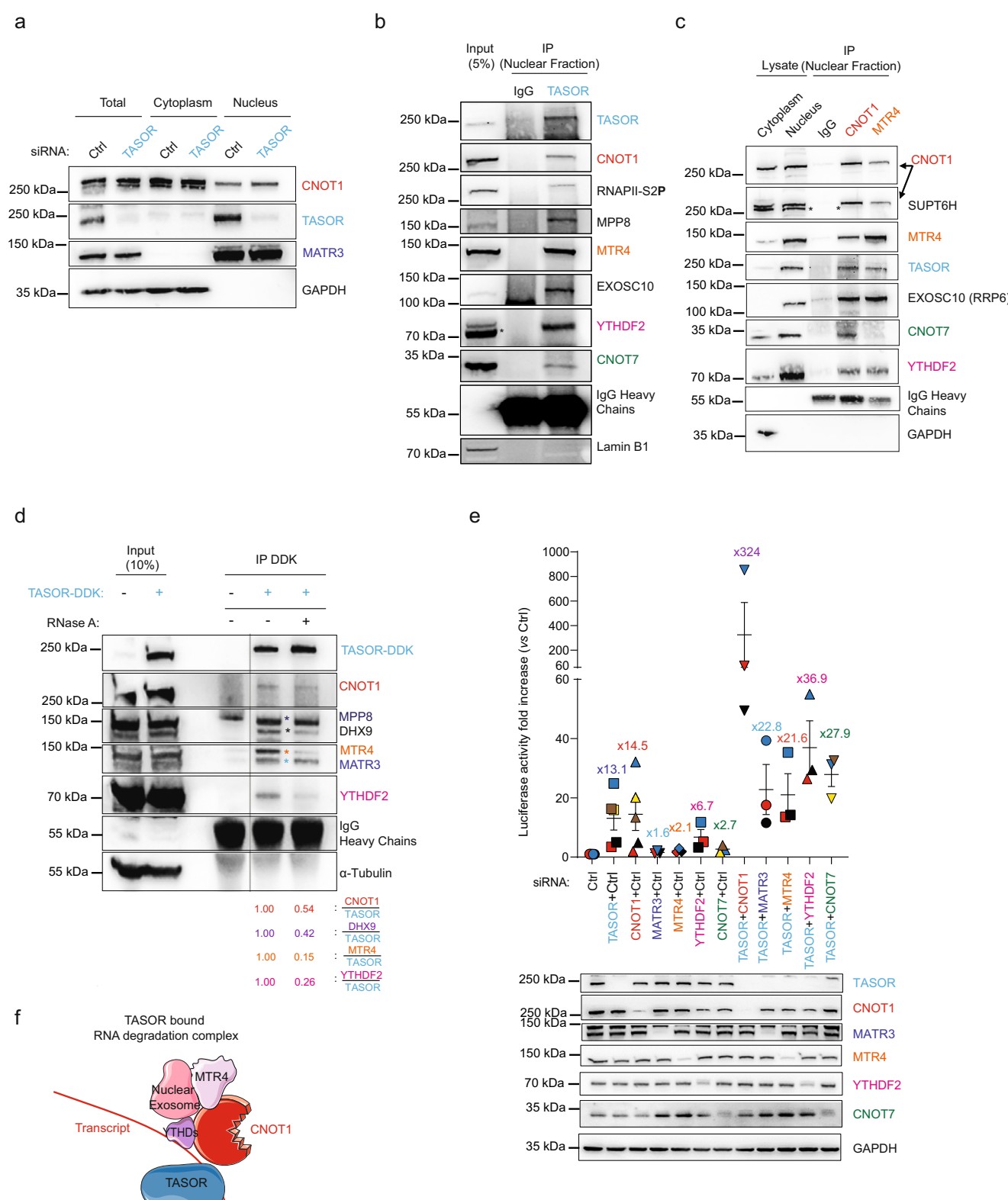

a TASOR isoform of 1512 amino acids (NCBI Reference Sequence: NP_001106207.1). TASOR-ΔPARP construct was obtained by deleting the DNA sequence corresponding to the 106-319aa using the CloneAmp™ HiFi PCR Premix (639298-Takarabio) and the following 5′-3′-oriented primers: F-CCAGGAAGT ATGCAGTTGTGTCTTTTACTTACA, R-CTGCATACTTCCTGGGGATCTGA AAACTCC. pLentiCRISPRV2-sgTASOR-Cas9 was obtained by subcloning the following 5′-3′-oriented annealed primers: F-CACCGCTTTCCCAACTCGCATC CGT, R-AAACACGGATGCGAGTTGGGAAAGC), containing the sgRNAs

targeting the first exon of TASOR, with the enzyme BsmBI. For complementation assays, pLenti-TASOR-DDK was made resistant to the guide by mutating the sgRNA-targeted sequence with 5′-CCAACAGACGCCTCGTGGGAGTCA-3′. The TASOR ORF was then subcloned into the pAS1B-HA vector. DDK-CNOT7 plasmid is a kind gift from Nancy Standart. The Dox-inducible miR CNOT1 and miR Luc pINDUCER10 plasmids were kind gifts from Alfonso Rodriguez-Gil. The R42A mutant of Vpx HIV-2 Ghana was produced by site-directed mutagenesis using the pAS1B-HA-Vpx HIV-2 Ghana construct as a template.

**Fig. 6 TASOR interacts and cooperates with nuclear RNA degradation factors. a** TASOR is a nuclear protein. Cytoplasmic and nuclear protein extracts from HeLa HIV-1 LTR-ΔTAR-Luc cells were loaded on a SDS-PAGE gel. GAPDH and MATR3 are markers of the cytoplasmic and nuclear fractions respectively (n > 3). **b** Endogenous TASOR interacts with the endogenous, nuclear CNOT1, and its partners CNOT7, YTHDF2, the endogenous TRAMP-like/NEXT/PAXT component MTR4, and the endogenous RNA exosome factor EXOSC10. HeLa HIV-1 LTR-ΔTAR-Luc cells were fractionated and endogenous TASOR immunoprecipitation was performed in the nuclear fraction. Lamin B1 is negative control (n > 3). **c** Reverse immunoprecipitation confirmed the interaction between endogenous CNOT1 and MTR4 proteins with endogenous TASOR in the nucleus. HeLa HIV-1 LTR-ΔTAR-Luc cells were fractionated and endogenous CNOT1 and MTR4 immunoprecipitation was performed in the nuclear fraction. The asterisk shows the SUPT6H band (n = 3). **d** TASOR interacts with the known CNOT1 partners in an RNA-dependent manner. DDK and TASOR-DDK vectors were transfected in HeLa HIV-1 LTR-ΔTAR-Luc cells. After 48 h, lysates were treated or not with RNase A for 30 min at room temperature. Anti-DDK immunoprecipitation was then performed. All lanes are from the same gel (n > 3). **e** TASOR cooperates with the nuclear RNA destabilization/degradation factors. After 72 h of siRNA transfections in HeLa HIV-1 LTR-ΔTAR-Luc, cells were lysed and luciferase activity was measured and normalized on protein concentration (n = 5 for the siRNA TASOR and siRNA CNOT1 conditions; n = 3 for other siRNA transfections; each color represents one different independent experiment, mean and SEM are shown). **f** Schematic representation of RNA-mediated interactions between TASOR and CNOT1 and its partners: the TRAMP-like/NEXT/PAXT component MTR4, the Nuclear Exosome, and the m$^6$A reader YTHDF2. Source data are provided as a Source data file.

**Virus and VLP production**. VSV-G pseudo-typed viruses and VLPs were produced in 293 T by the calcium-phosphate co-precipitation method. SIV3 + ΔVprΔVpx packaging vectors were a gift from N. Landau and is described in[66]. VLPs (Vpx expressed from pAS1B and incorporated into VLPs) were obtained from co-transfection of VSV-G plasmid, SIV3 + ΔVprΔVpx packaging vector, and pAS1B-HA-Vpx HIV-2 Ghana (WT or R42A) or pAS1B-HA (empty). The HIV-1 LTR-EGFP virus presented in Fig. 4a was a gift from Jeremy Luban (Addgene plasmid # 115809—and already described in ref. [13]) and was VSVg pseudotyped. Cell culture medium was collected 48 h after transfection and filtered through 0.45 μm pore filters. Viral particles or VLPs by sucrose gradient and ultra-centrifugation. The incorporation of Vpx into VLPs was assessed by western blot.

**Cell lines**. Cell lines were regularly tested for mycoplasma contamination: contaminated cells were discarded to perform experiments. ATCC-purchased HeLa (CCL-2), HEK293T (CRL-3216), THP-1 (TIB-202), and Jurkat (TIB-152) cells were cultured in media from ThermoFisher: DMEM (HeLa, 293T), RPMI (THP1 monocytes, Jurkat cells) containing 10% heat-inactivated fetal bovine serum (FBS, Dominique Dutscher), 1000 units/mL penicillin, and 1000 μg/mL streptomycin. HeLa LTR-ΔTAR-Luc cells were generated in the laboratory of Stephane Emiliani from the HeLa LTR-Luc cells described by du Chené et al.[67]. J-Lat A1 miR Luc and miR CNOT1 cells were produced by transductions of VLPs-containing pINDU-CER10-miR Luc and pINDUCER10-miR CNOT1 respectively and cultured for 4 days prior to puromycin selection. The Jurkat cell lines were infected with HIV-1-EGFP (Yurkovetskiy 2018) and then the EGFP positive cells were sorted by flow cytometry with the BD FACS ARIA3 cytometer of the CYBIO platform (Institut Cochin). After 2 weeks of culture, a second sorting was performed to recover the cells that had become EGFP negative. These latently infected cells were then transduced with VLPs containing the doxycycline-inducible miRNA Luc or CNOT1 construct (pInducer10). Puromycin was added 4 days after transduction to purify cells that stably express miR Luc or miR CNOT1 upon doxycycline treatment.

**siRNA treatment**. siRNA transfections were performed with DharmaFECT1 (Dharmacon, GE Lifesciences). The final concentration for all siRNA was 100 nM. The following siRNAs were purchased from Sigma Aldrich: siTASOR: SASI_Hs02_00325516; siCNOT1: SASI_Hs02_00349201; siCNOT7: SASI_Hs02_00344676; siDICER: SASI_Hs01_00160748 siMATR3: SASI_Hs02_00352277; siMTR4: SASI_Hs01_00072261 siYTHDF2 SASI_Hs01_00133218; siEXOSC10: SASI_Hs01_00183400. The non-targeting control siRNAs (MISSION siRNA Universal Negative Control #1, SIC001) were purchased from Sigma Aldrich.

**Luciferase activity assay**. Cells were washed twice with phosphate-buffered saline (PBS) then lysed directly in wells using 1× cell culture lysis reagent (Promega). Cell lysates were clarified by centrifugation, luciferase activity was measured using a luciferase assay system (Promega) and a TECAN multimode reader Infinite F200 Pro and data were normalized on protein concentration with the use of Pierce BCA Protein Assay Kit (23225-ThermoFisher).

**Flow cytometric analyses**. TNFα-treated (1 ng/ml) or untreated cells were collected and resuspended in PBS-EDTA (0.5 mM). Data were collected and analyzed with a BD Accuri C6 cytometer or with a BD LSRFortessa Cell Analyzer and software CFlow Plus or with FlowJo V10. At least 10,000 events in P1 were collected, the GFP-positive population was determined using a GFP-negative population when possible or arbitrary (as for J-Lat cells), and the same gate was maintained for all conditions. Analysis was performed on the whole GFP-positive population.

**Nuclear Run On (NRO)**. HeLa LTR-ΔTAR-Luc cells were grown in 10 cm dishes and transfected with siRNA ctrl, TASOR + ctrl, CNOT1 + ctrl, and TASOR + CNOT1 at a final concentration of 100 nM each. NRO was performed as precisely described by Roberts and colleagues[68], except for these specific points: At step 14, RNA extractions with rDNase treatment were performed with NucleoSpin RNA, Mini kit (740955.250, Macherey-Nagel). At Step 44, Reverse transcription was performed with Maxima First Strand cDNA Synthesis Kit with dsDNase (K1672, ThermoFisher). qPCR was finally performed as described in the RT-qPCR section.

**RT-qPCR**. Except for step 14 in the *Nuclear Run On* experiments, all RNA extractions and purifications were based on a classical TRI Reagent (T9424-200ML, Merck) protocol. Reverse transcription steps were performed with Maxima First Strand cDNA Synthesis Kit with dsDNase. PCRs were performed thanks to a LightCycler480 (Roche) using a mix of 1x LightCycler 480 SYBR Green I Master (Roche) and 0.5 μM primers which sequences are described in Supplementary Table 2.

**FAIRE-qPCR**. Around 6×10⁶ HeLa HIV-1 LTR-Luc cells were transfected with siRNA Ctrl or siRNA TASOR for 72 h. TNFα (10 ng/mL) was added or not 4 h before cell recovery. The FAIRE-qPCR protocol from[69] was then followed. At step 3.2, the chromatin of each sample was sonicated with a Diagenode BIORUPTOR Pico with 10 rounds of sonication 30 s on/30 s off. qPCR was performed using the primers listed in Supplementary Table 2.

**Cell fractionation**. HeLa cells grown in 10 cm dishes were washed with cold Dulbecco's PBS 1× (ThermoFisher). After trypsinization (25200056, Thermo-Fisher), cells were recovered in 1.5 mL tubes and washed once with ice-cold PBS. After 4 min of centrifugation at 400 × g, 500 μL of Cytoplasmic Lysis Buffer (10 mM TRIS-HCl pH7.5, 10 mM NaCl, 3 mM MgCl₂, and 0.5% IGEPAL® CA-630 (I8896-100ML-Merck)) was added on the cell pellet and resuspended pellet was incubated on ice for 5 min. Cells were then centrifuged at 300 × g for 4 min at 4 °C. The supernatant was saved for cytoplasmic fraction. The pellet was resuspended with 1 mL of Cytoplasmic Lysis Buffer and re-centrifuged at 300 × g for 4 min at 4 °C twice. Finally, the nuclear pellet was lysed with 200 μL of RIPA Buffer.

**Immunoprecipitation, western blot procedures, and antibodies**. For TASOR-DDK or DDK-CNOT7-immunoprecipitations: HeLa/293T cells grown in 10 cm dishes were transfected with pLenti-DDK or with pLenti-TASOR-DDK or Flag-CNOT7 plasmids with CaCl₂. 48 h after transfection, cells were lysed in 500 μL RIPA buffer (50 mM Tris-HCl pH7.5, 150 mM NaCl, 1 mM MgCl₂, 1 mM EDTA, 0.5% Triton X100) containing an anti-protease cocktail (A32965, ThermoFisher). Cell lysates were clarified by centrifugation and a minimum of 200 μg of lysate was incubated with pre-washed EZview™ Red ANTI-FLAG® M2 Affinity Gel (F2426, Merck) at 4 °C, under overnight rotation. After three washes in wash buffer (50 mM Tris-HCl pH7.5, 150 mM NaCl), immunocomplexes were eluted with Laemmli buffer 1× and were separated by sodium dodecyl sulfate-polyacrylamide gel electrophoresis (SDS–PAGE). For endogenous Immunoprecipitation, the same procedures were followed but a minimum of 400 μg of lysate was used for overnight IP with 4 μg of the corresponding antibody and pre-washed Pierce™ Protein A/G Magnetic Beads (88802, ThermoFisher) were added for 1 h at room temperature. Following transfer onto PVDF membranes, proteins were revealed by immunoblot. Signals were acquired with Fusion FX (Vilber). The following antibodies, with their respective dilutions in 5% skimmed milk in PBS-Tween 0.1%, were used: anti-Flag M2 (F1804-200UG, lot SLCD3990, Merck) 1/1000; anti-TASOR (HPA006735, lots A106822, C119001, Merck) 1/1000—for IF assays: 1/500; anti-TASOR (HPA017142, Merck) for Immunoprecipitation experiments, anti-MPP8 (HPA040035, lot R38302, Merck) 1/500; anti-CNOT1 (For WB: 66507-1-Ig, Proteintech, 1/1000; For IP: 14276-1-AP, Proteintech); anti-CNOT7 (14102-

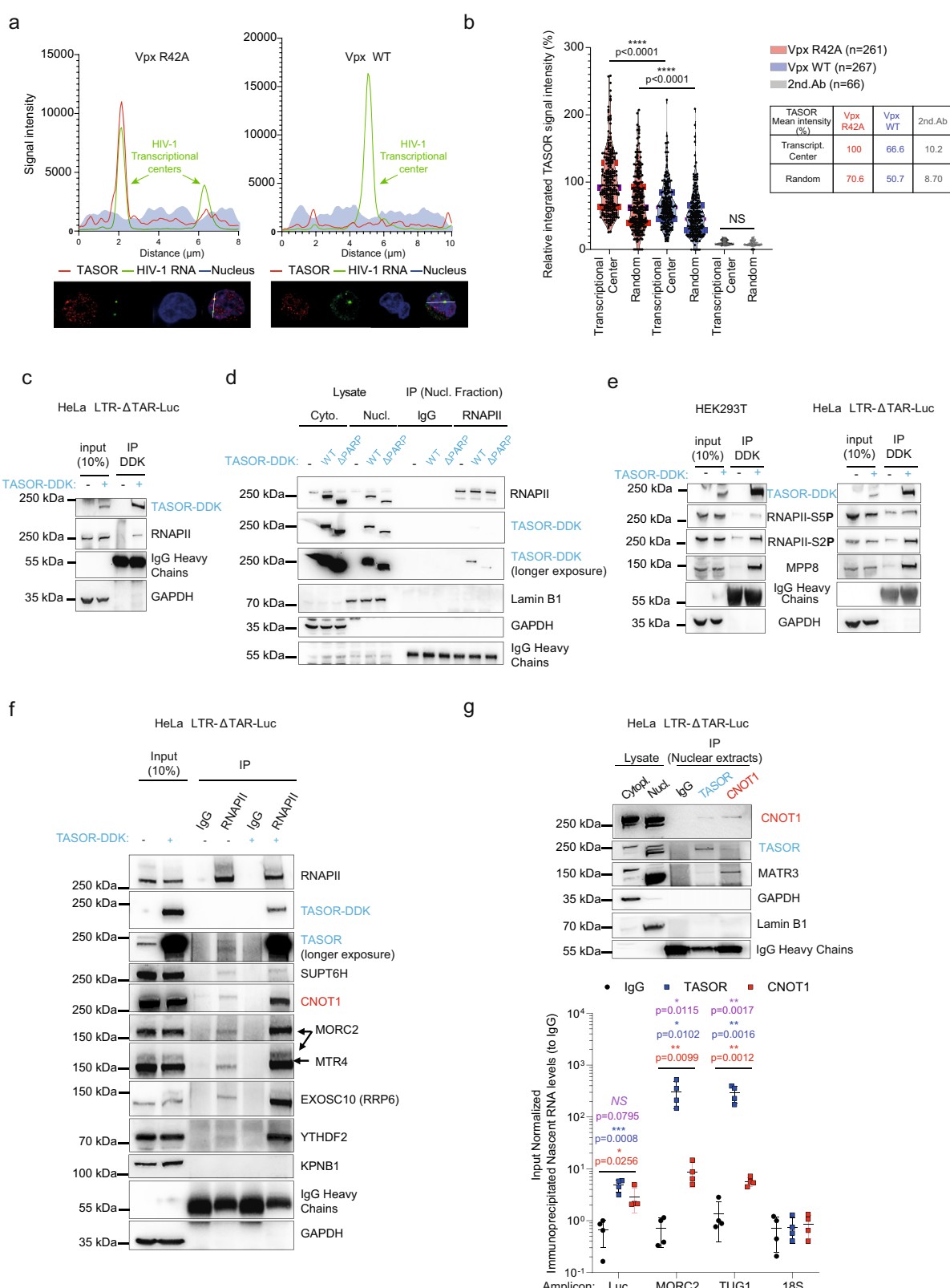

1-AP, Proteintech) 1/500; anti-CNOT9 (22503-1-AP, Proteintech) 1/500; anti-DHX9 (17721-1-AP, Proteintech) 1/1000; anti-EXOSC10 (11178-1-AP, Proteintech) 1/1000; anti-HLTF (ab17984, Abcam) 1/1000; anti-IGF2BP1 (22803-1-AP, Proteintech) 1/1000; anti-KPNB1 (HPA029878-100ul, lot D114771, Merck) 1/1000; anti-MATR3 (12202-2-AP, Proteintech) 1/1000; anti-MORC2 (PA5-51172, TermoFisher) 1/1000; anti-MTR4 (For WB and IP: 12719-2-AP, Proteintech) 1/1000; anti-PPHLN-1 (HPA038902, Lot A104626, Merck) 1/1000; anti-RNAPII (For IP and WB: F12, sc-55492, Santa Cruz Biotechnology) 1/1000; anti-Ser2P-

RNAPII (13499S, Cell Signaling technology) 1/1000 in 2.5% BSA-TBS-Tween 0.1%; anti-Ser5P-RNAPII (13523S, Cell Signaling technology) 1/1000 in 2.5% BSA-TBS-Tween 0.1%; anti-SUPT6H (For WB and IP: 23073-1-AP, Proteintech) 1/1000; anti-U1snRNP70 (sc-390899 (C3), Santa Cruz Biotechnology) 1/500; anti-YTHDF2 (24744-1-AP, Proteintech) 1/1000; anti-ZFC3H1 (HPA007151, Merck) 1/1000, anti-β-Actin (AC40, A3853, Merck) 1/1000; anti-αTubulin (T9026-.2mL, lot 081M4861, Merck) 1/1000; anti-GAPDH (6C5, SC-32233, Santa Cruz) 1/1000. All secondary antibodies anti-mouse (31430, lot VF297958, ThermoFisher) and

**Fig. 7 TASOR recruits RNA-degradation factors onto elongating RNAPII to silence gene expression. a** TASOR colocalizes with active HIV-1 transcriptional centers in HIV-1-infected Jurkat cells. Latently HIV-1-EGFP-infected Jurkat cells were transduced with VLPs containing Vpx R42A or WT for 24 h and treated with TNF (1 ng/mL). LTR-driven unspliced (us) RNAs were marked with *egfp* probes and TASOR by immunofluorescence. The graph corresponds to the signal intensities across the white line. **b** Vpx WT triggers a significant loss of TASOR at the HIV-1 transcriptional centers. TASOR signal at HIV-1 transcription centers in the Vpx R42A delivery condition was set as 100%. ($n = 261$, $n = 267$, and $n = 66$ for the Vpx R42A, Vpx WT, and secondary antibody (2nd.ab) Alexa 594 conditions, respectively; two-sided unpaired $t$ test was applied: ****$p < 0.001$). **c** TASOR interacts with endogenous RNAPII in HeLa HIV-1 LTR-ΔTAR-Luc cells. GAPDH is a negative control ($n > 3$). **d** TASOR affinity for RNAPII is greatly reduced upon depletion of the N-terminal PARP domain. Lamin B1 and GAPDH are negative controls ($n = 3$). **e** TASOR interacts with elongating RNAPII in HEK293T and HeLa HIV-1 LTR-ΔTAR-Luc cells. Ser5-phosphorylated and Ser2-phosphorylated RNAPII are markers of RNAPII in the initiating and elongating phases of transcription, respectively, MPP8 is a positive control. GAPDH is a negative control ($n > 3$). **f** TASOR recruits CNOT1 and its partners onto RNAPII. DDK and TASOR-DDK vectors were transfected in HeLa HIV-1 LTR-ΔTAR-Luc cells. TASOR was revealed either with an anti-DDK (TASOR-DDK) or with an anti-TASOR to detect the endogenous protein. SUPT6H is a control of a known RNAPII partner. MORC2 is necessary for HUSH-mediated gene silencing according to ref. [12]. GAPDH is a negative control ($n = 3$). **g** TASOR and CNOT1 are found in complex with the HIV-1 nascent transcript. Nascent RNAs from HeLa HIV-1 LTR-ΔTAR-Luc cells were labeled and anti-TASOR and anti-CNOT1 immunoprecipitations were performed from the nuclear extract. GAPDH and TASOR partner MATR3 are markers of the cytoplasmic and nuclear fractions, respectively. Immunoprecipitated RNAs were normalized on input signal ($n = 4$; mean and SEM are shown; two-sided unpaired $t$ test was applied: purple stars or NS indicate statistical differences between TASOR and CNOT1 conditions; blue stars: TASOR vs IgG, red stars: CNOT1 vs IgG). Source data are provided as a Source data file.

anti-rabbit (31460, lots VC297287, UK293475 ThermoFisher) were used at a 1/10,000 dilution before reaction with Immobilon Forte Western HRP substrate (WBLUF0100, Merck Millipore).

**Nascent RNA IP**. HeLa LTR-ΔTAR-Luc were grown in 10 cm dishes. Forty-eight hours post-transfections, the cells were recovered and labeling of nascent RNAs was undertaken with the *Nuclear Run On* protocol[68]. 10% of nuclei were put aside for purification of BrUTP incorporated Nascent RNAs (input). From the remaining nuclei, RNA-IP was then undertaken with the MagnaRIP Magna RIP™ RNA-Binding Protein Immunoprecipitation Kit protocol (17-700, Merck) from step I.3. TASOR, CNOT1 or anti-Rabbit IgG immunoprecipitations were performed with 4 µg of anti-TASOR (HPA017142- Merck), anti-CNOT1 (14276-1-AP- Proteintech), anti-Rabbit IgG (12–370) on Nuclei Lysates at 4 °C overnight. From step III.6 of the MagnaRIP protocol, 10% of IPed lysate was recovered and washed three times with IP wash buffer (50 mM Tris-HCl pH 7.5, 150 mM NaCl) and loaded on SDS-PAGE gel to check DDK pull-down efficiency. The remaining IPed Lysate was then washed from step III.6 of the MagnaRIP protocol. Finally, purification of BrUTP incorporated IPed Nascent RNAs was undertaken from step 25 of the *Nuclear Run On* protocol[68]. After purification, RT-qPCR of all purified RNAs was performed with primers listed in Supplementary Table 2.

**RNA-seq**. Six hundred thousand HeLa cells containing an integrated copy of LTR-HIV-1-Luciferase Firefly were plated in 6-well plates. Cells were transfected with siRNAs targeting Mock, or TASOR or CNOT1 or both TASOR and CNOT1 expressions using Dharmafect 1 (Horizon Discovery). 72 h after transfection, cells are washed twice with PBS. A 100 µL aliquot of the pipetted cells is collected for protein assay, and monitoring of TASOR and CNOT1 silencing by western blot; the cells are then centrifuged and lysed with 500 µL TRI Reagent. RNA extraction was performed using a standard TRIZOL/Phenol/Chloroform protocol. RNAs resolubilized in 20 µL of RNase Free water were then passed through an RNA purification column (from the Macherey Nagel Ref:740955.250 kit) and subjected to rDNAse digestion from the purification kit. The eluted total RNAs are submitted to the Bioanalyzer for quality control at the Genom'IC platform, Institut Cochin. After quality control analysis, 1.5 µg of RNA from each sample was sent for sequencing. Libraries were prepared using the NEB Ultra II protocol (stranded polyA+ RNA-seq) and sequenced with an Illumina NextSeq 500 as 2 × 75 bp paired-ends (PE) at the Genom'IC platform. Raw and processed RNA-seq data were submitted to GEO access number GSE184399.

Trimmed PE reads were mapped against the human reference genome, hg38 (Gencode and gene annotation version 29) using STAR (version 2.7.5c)[70]. The parameters used for the mapping were as follows:–outFilterMultimapNmax 1000 (1000 alignments allowed per read-pair),–alignSJoverhangMin 8 (minimum overhang for unannotated junctions). Read counting was performed using TEtranscripts and TElocal (version 1.1.1) (https://github.com/mhammell-laboratory/TElocal) from the TEToolkit suite[71] with recommended parameters. DESeq2 (version 1.30.1)[72] was used for the differential expression analysis of genes or transposable elements.

TASOR and CNOT1 interactions on gene expression were modeled with a contrast matrix in DESeq2 using the following design: ~"siTASOR+siCtrl" + "siCNOT1+siCtrl" + "siTASOR+siCtrl":"siCNOT1+siCtrl". Only genes upregulated upon double silencing of TASOR and CNOT1, and with expression levels >1 TPM, were considered in the analysis. The log2 fold change (FC) values obtained represents the difference: log2(FC(siCNOT1+siCtrl vs siCtrl)) − log2(FC(siTASOR+siCNOT1 vs siTASOR+siCtrl)). Thus, genes with a log2(FC) > 0 and an adjusted $p$

value < 0.05 (Wald test after Benjamini–Hochberg multiple testing correction) were considered as resulting from a synergistic effect (positive interaction) of TASOR and CNOT1, genes with a log2(FC) < 0 and an adjusted $p$ value < 0.05 from a negative synergistic effect (negative interaction) and genes with an adjusted $p$ value > 0.05 from an additive effect (or additive interaction).

**smRNA FISH**. Probes were developed using the designer tool from http://singlemoleculefish.com/. A unique set of probes was designed to detect the *egfp* region in HIV-1 LTR-*egfp*. The set contained 25 probes, each probe was 18 nt long, using a masking level of 3–5, and at least 2 bp spacing between single probes.

Probes (3′–5′ orientation): #1 accacagttttcctcttc; #2 gacaagtggcctcaccag; #3 acctgcctctacagttac; #4 gtgttcaagtcgcactcg; #5 ccgttcgactgagactt; #6 taaacgtggtgtcccttc; #7 ggttgggaccagtgatgg; #8 tgtatgccgcaggtcaca; #9 tcgtctataggtctggtg; #10 aatttaggcggtacgggc; #11 cgtccttgcctgttagaa; #12 cgactccacttcaagctt; #13 tctgtgggaccagttgtc; #14 gctcgacttcccgtaact; #15 agtaccgtctattcgtct; #16 ttgccgtaattccagtta; #17 ttataacttctgccctcg; #18 cgtcgacccgctagtaat; #19 cgtcttgtgtgggatagcc; #20 gcacgacgacggactatt; #21 tgagtctcacgggacagt; #22 tgggtttgctctttgctc; #23 ccacgacgacctcaaaca; #24 gcgacgaccttagtgtga; #25 cgtacctactcgacatgt.

Probes were conjugated with Quasar 670. VLPs containing Vpx WT or VPX R42A or mock have been delivered on Jurkat cells latently infected with HIV-1 LTR-EGFP for 24 h. Cells were treated with 1 ng/ml TNFa (Sigma, catalog T0157) 16 h before fixation. The cells were washed with 10 mL of PBS solution and then immobilized on Cell-Tak coated six-well plates. Cells were then fixed with 5% formaldehyde in PBS for 10 min. Fixed cells were stored in 70% EtOH at 4 °C for a minimum of 1 h to permeabilize the cell membranes. Probes were diluted to a final concentration of 50 nM in 1 g/mL dextran sulfate, 2xSSC, and 10% formamide and allowed to hybridize at 37 °C overnight in a dark and humid chamber. Wash steps and DAPI staining were performed as described in Hansen et al.[73]. Then, cells were permeabilized once more with PBS-Triton 0.1% for 10 min at room temperature for TASOR immunofluorescence staining. After PBS-mediated washes, a 1/500 anti-TASOR (HPA006735-Merck) dilution in IF buffer (PBS-Tween 0.05%-BSA 0.2%) was incubated on the cells for 1 h at RT in a dark chamber. The cells were then washed three times with PBS and anti-rabbit Alexa 594 was added at a dilution of 1/1000 in IF buffer for 1 h at room temperature in a dark chamber. After three PBS-mediated washes, the coverslips were mounted on slides with the use of ProLong™ Diamond Antifade Mountant (P36970-ThermoFisher). Imaging was performed with a Spinning-Disk IXplore Olympus confocal microscope, using a ×100 objective, a Hamamatsu sCMOS Orca flash 4.0 V3 camera, and the cellSens Dimension acquisition software. GFP-positive cells were randomly captured. Analysis of images was performed with ImageJ and with the help of the Python core analysis package big-fish available at https://fish-quant.github.io/.

**Reporting summary**. Further information on research design is available in the Nature Research Reporting Summary linked to this article.

## Data availability

The RNA sequencing data generated in this study have been deposited in the GEO database under the accession code GSE184399. The CUT&Tag and CUT&RUN data of TASOR GSM4710610 and H3K9me3 GSM4710590 and GSM4710594, respectively, were published by Douse et al.[12] and deposited at GEO under the accession number GSE155693. ChIPseq and RNAseq were performed by Liu et al.[55] and deposited at GEO under the accession number GSE95374. Source data and full western blots are provided with this article.

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

## Acknowledgements

We thank all the Retrovirus, Infection and Latency group members at Institut Cochin for fruitful discussions. We warmly thank Nancy Standart for providing the Flag-CNOT7 plasmid and Alfonso Rodriguez-Gil for sharing the pINDUCER10 miR Luc and miR CNOT1 vectors. We thank Adham Safieddine for sharing his smRNA FISH protocol. We acknowledge the CYBIO, GENOM'IC, and IMAG'IC platforms of the Institut Cochin. This work was supported by grants from the "Agence Nationale de la Recherche sur le SIDA et les hépatites virales" (ANRS) and SIDACTION. R.M. was supported by ANRS and SIDACTION. B.M. is supported by ENS Paris Saclay and P.L. by Université de Paris.

Work in the laboratory of G.C. is supported by grants from Fondation pour la Recherche Médicale (FRM, DEQ20180339170), the Agence Nationale de la Recherche (LABEX SIGNALIFE, ANR-11-LABX-0028-01; RetroMet, ANR-16-CE12-0020; ImpacTE, ANR-19-CE12-0032), and CNRS (GDR 3546).

## Author contributions

R.M. and F.M.-G. conceived the study, designed experiments, and interpreted data. R.M., M.M., P.L., and B.M. performed experiments. B.M. and M.H. brought their expertise and tools for smRNA FISH experiments. B.M. and R.M. analyzed the smFISH data. F.B. analyzed previously published ChIPseq, CUT&Tag, and RNAseq data and with R.M. compared them with our RNAseq data. S.L. and G.C. analyzed RNAseq data, including the expression of cellular genes and retroelements. V.V. contributed to the set-up of TASOR overexpression experiments in a CRISPR-TASOR background. S.E. and S.G.-M. provided critical reagents and brought their expertise through regular discussions. S.G.-M. analyzed data and performed splicing experiments. R.M. and F.M.-G. wrote the paper. All authors reviewed and edited the manuscript.

## Competing interests

The authors declare no competing interests.
