## [Peer Review File · Nature Communications]

TASOR epigenetic repressor cooperates with a CNOT1 RNA degradation pathway to repress HIVReviewers' Comments:

Reviewer #1:

Remarks to the Author:

TASOR epigenetic repressor cooperates with a CNOT1 RNA degradation pathway to repress HIV
Matkovic et al.

In this study, the authors seek to identify new binding partners of the HUSH complex to gain some mechanistic insight into how the HUSH complex and associated complexes may repress HIV. In a yeast-2-hybrid screen, the authors identify CNOT1 as a novel interaction partner of the HUSH complex component TASOR. The HUSH complex exerts some homology with the yeast RNA-induced silencing (RITS) complex, due to it possessing some of the same domains. Like the RITS complex, TASOR interacts with MTR4 and the exosome, which are binding partners of CNOT1. The RITS complex is associated with elongating RNA polymerase II and with small RNA-induced deposition of repressive H3K9me3 at retroelements. The interaction of TASOR with CNOT1, MTR4 and the nuclear exosome is dependent on RNA and TASOR interacts with elongating RNA pol II and the nascent HIV RNA reporter transcript. Interestingly, co-depletion of TASOR and CNOT1 or other associated RNA degradation factors leads to a synergistic increase in Luciferase production from the HIV reporter. Depletion of DICER, in contrast, has no effect on HIV reporter transcripts. The authors conclude that HUSH represses HIV at the transcriptional and post-transcriptional levels through HUSH and the CNOT1 RNA degradation pathway. This study is novel, generally well-performed and interesting but some points need further clarification.

Major comments

1. Figure 1b: The TASOR-induced repression here is not remarkable (2-fold). It would be more convincing if this repression could be increased by increasing overexpression of TASOR. An additional negative control of GFP or other overexpressed construct should also be included.
2. Figure 1cd: TASOR depletion does not affect total RNA (2.6-fold increase) much more than it affects nascent RNA (1.3-fold increase). In contrast, it appears to greatly affect protein levels (18x in figure 1c). This should be verified with RNA and protein measured in the same experiments and discussed (presumably TASOR tethers RNA to chromatin and its depletion causes RNA to be re-localized to the cytoplasm).
3. It is not clear if TASOR and CNOT1 can inhibit the HIV replication cycle (exerting transcriptional and post-transcriptional blocks) and / or play a role in repressing latent HIV or if these complexes only act on transgenes, especially since TASOR seems to mainly affect chromatin accessibility across the Luciferase transgene (figure 4de). It would be interesting to have some insight into this question using WT HIV.

Minor comments

1. Figure 1b: Lenti-DDK and Lenti-TASOR-DDK need to be defined in the legend. The TASOR band should be labelled as TASOR not DDK.
2. Figure 1e: A Vpx mutant was used as a negative control that does not bind and induce degradation of TASOR. This control should be included in the figure rather than in supplementary data.
3. Figure 2b: It would be clearer if 'DDK' was re-labelled as 'TASOR' on the IP or TASOR cDNA to distinguish from endogenous TASOR. This is true for many figures including figure 4fg.
4. In the first sentence of the abstract, LINE-1s, retrotransposons and retroelements are mentioned. This is confusing because LINE-1 elements are retrotransposons and retroelements.
5. Figure 2efg: Independent experiments are represented by different colours, which makes the figure

unclear. It would be easier to represent different treatment groups by different colours.

6. Figure 3d: Can the authors comment on why there is so much variability in this figure?

7. Figure 4a: It would be helpful to show on this diagram that the MORC2 and TUG1 genes do not overlap. This could be shown by staggering the UCSC gene maps so that they are not simply a continuous line.

8. Figure 4g: I am not sure if it is necessary to have a split scale on the y axis. This graph would benefit from a simpler scale.

9. Please revise the following unclear sentence: 'However, HIV-1 integration in poorly transcribed, desert, geneless, or centromeric regions can result in the repression of HIV-1 expression due to multiple epigenetic mechanisms'.

10. Please revise the following sentence as the word 'repressive' appears twice: 'The HUSH complex - TASOR, MPP8 and periphilin- was identified as a potential player in HIV repression, propagating repressive H3K9 repressive marks in a position-effect variegation manner in a HIV-1 model of latency.'

11. The text is unclear in parts and would benefit from some editing/re-writing and grammar corrections to aid readability.

12. A structure prediction analysis of TASOR is mentioned: 'Structure/function prediction analyses of TASOR first 900 amino acid (aa) sequence using the PSIPRED server (UCL Bioinformatics group) revealed a PARP13-like PARP domain at its N-terminus region (300 aa) with high probable and reliable functions in mRNA binding, splicing and processing, together with the SPOC (Spen paralog and ortholog C-terminal) domain, often associated with transcription repression'. Here, the authors should cite and refer to the recently published article (PMID: 33009411), in which structural details of TASOR, including its pseudo-PARP domain and the similarity between HUSH and the RITS complex are discussed.

13. CNOT1 should be named in full at first mention and it would be helpful to state its cellular localization.

14. The authors should discuss why they couldn't efficiently overexpress CNOT1.

15. Please revise 'when inhibiting both proteins expression' to 'when depleting both proteins' or similar.

16. Please revise the following unclear section: 'Because we could not efficiently overexpress CNOT1, we chose the deadenylase CNOT7 partner to pull-down the complex in the reverse situation. We immunoprecipitated the DDK-tagged CNOT7 and revealed an interaction with endogenous TASOR, and the other CCR4-NOT complex components CNOT9 and CNOT1 proteins (Fig. 2d)'.

17. All Westerns should have kDa size markers.

Reviewer #2:

Remarks to the Author:

Authors have previously reported in a very nice study in Nature Microbiology that TASOR is an important repressor of HIV expression, involving H3K9me3 histone mark, countered by the HIV-2

protein vpx. Matkovic et al. now describe a new function of TASOR in mRNA processing co/post-transcriptionally. Authors find new partners of TASOR which seem to cooperate in order to repress gene expression from an HIV-LTR/luciferase model. This complex is reminiscent of the yeast RITS complex. These findings are novel, in a well conducted study, and extend our understanding of how TASOR can repress expression of HIV at the co/post-transcriptional level. This study would however benefit from the following revisions.

Major points :

- 1) Authors should at some point use a better HIV model (full genome) to confirm that their new partners play an important role in a more physiological context. Although primary cells would be ideal, authors could also use the J-lat model they previously employed in their Nat. Microb.
- 2) IPs are not always sufficiently confirmed using endogenous proteins. Indeed, interactions are first assessed in Figure 2 using over-expressed proteins and both Y2H and over-expression experiments may yield false positives. However, authors later perform IPs using endogenous proteins that confirm some of their interactions. They should still confirm IP between TASOR and CNOT7 in their Figure 2 (either TASOR or CNOT7 IP would be sufficient). If this is not possible due to both antibodies not suitable for IP, authors could at least show WB of TASOR when TASOR-DDK is used and CNOT7 when CNOT7-DDK is used to compare them to endogenous levels (DDK alone). Same for figures 4b/c/f. There is no IP of endogenous proteins for phosphorylated forms of RNAPII. Also for RMAPII/TASOR interaction, there is endogenous IP in figure f, authors should show a stronger exposure of their blots to better assess endogenous TASOR presence after RNAPII IP ? Same for YTHDF2 and CNOT1 which seem absent unless there is overexpression. The fact that over-expression of the protein brings its associated partners is a good thing but endogenous proof is important.
- 3) An experiment of RNA-IP was performed at the end. This is a very nice experiment and suggests a direct effect of the TASOR. Authors could IP other proteins that interact with TASOR and probably function together to better confirm their model (e.g. at least one of the factors that have the best synergistic effect such as CNOT1). IPs against endogenous TASOR and/or some endogenous proteins (not only DDK-protein overexpression) are required (at least one protein of the complex should be studied using endogenous IP).

Minor points :

Line 124 : « between ... and » not « with »

Line 156 : I believe this experiment shows that both TASOR and CNOT complex can inhibit LTR driven luciferase mRNA, but do not formally demonstrate collaboration. At this point, we don't know if interaction is mandatory for the observed effect. I would change the phrasing accordingly (e.g. strongly suggest instead of confirm).

Fig 3d : CNOT7 fold increase seems to be missing. Does that mean it's not significant ?

Reviewer #3:

Remarks to the Author:

Tasor epigenetic repressor cooperates with a CNOT1 RNA degradation pathway to repress HIV

Summary

In this study, Matkovic et al. propose that the Human Silencing Hub (HUSH) complex interacts with components of an RNA degradation pathway (CCR4-NOT) to repress HIV. The HUSH complex (TASOR, MPP8, and Periphilin) represses HIV provirus in addition to retroelements, retrotransposons and ribosomal DNAs through spreading of repressive epigenetic marks (H3K9me3). However, the precise mechanism of HUSH-mediated silencing is not clear. Using a protein-protein interaction approach, the authors found a physical interaction between TASOR and CNOT1, which is a scaffold of the RNA deadenylase CCR4-NOT complex. The authors then proposed that TASOR and CNOT1 interact in cells and "synergistically" repress HIV gene expression. In addition to CNOT1, TASOR appears to cooperate with MTR4 and the exosome, interacts with the RNA polymerase II elongating form and nascent

transcripts, and TASOR over-expression facilitates the association of RNA polymerase II with the RNA degradation complex. Overall, the study holds the potential for elucidating a previously undescribed functional link between HUSH and an RNA degradation pathway in gene regulation. However, in its current format it is found preliminary and thus further evidence in physiologically relevant models of disease are required to substantiate the proposed models.

Major concerns

1. In addition to the HIV provirus, HUSH regulates H3K9me3 spreading at human genomic domains including retrotransposons and retroelements. Providing evidence that HUSH-CCR4-NOT regulates these other host elements will broaden the scope of this work.
2. Loss of TASOR triggers the accumulation of HIV transcripts in a HeLa latency model, and it is proposed that the effect is post-transcriptional rather than transcriptional. However, long-term TASOR depletion can lead to indirect consequences impacting on HIV gene expression, but this alternative interpretation was not carefully considered.
3. In the abstract and throughout the text, the introduction that HIV-2 Vpx (which counteracts HUSH) mimics TASOR depletion is cumbersome and not well integrated in the entire story. It is known that Vpx induces HUSH degradation, but Vpx also has many accessory functions in the viral life cycle, thus it is unclear what events are HUSH-dependent and -independent Vpx functions. To assess this point, the authors could examine Vpx functions in the context of a relevant latency model with and without TASOR.
4. Much of the work builds on a non-physiological relevant model in HeLa cells containing an LTR-driven luciferase reporter in which the TAR stem-loop has been deleted (Figure 1). This molecular event does not occur in replication-competent proviruses, but only under special circumstances in proviruses that undergo latency (e.g., work by Verdin lab). Also, HeLa are not relevant to HIV infection and the observed phenotype by which TAR deletion enhances HIV gene expression is artificial. Physiologically relevant systems (e.g., CD4+ T cells) must be put into work to define how HUSH could inhibit the turnover of HIV transcripts.
5. Throughout the text, overexpression of proteins has been used to propose models of HIV gene regulation (Figure 1), but it is unclear what is the physiologic relevance of these artificial experiment in HeLa cells.
6. An interaction between HUSH and CCR4-NOT must be shown in physiologically relevant systems and not under overexpression conditions (Figure 2).
7. The authors utilized publicly available ChIP-seq and RNA-seq datasets (Liu et al.) to assess factor and epigenetic mark occupancy, but it was unclear in the text in what cell types these were collected and whether they hold physiologic relevance.

Minor concerns

1. The text needs grammar and scientific improvement. Example on pg. 3. Vpx does not “degrade” TASOR-MPP8 because it is not an E3 ligase itself, but it is needed for “TASOR-MPP8” degradation. Likewise, TNF is not a “transcriptional activator” per se, but a known ligand that induces a signaling and transcriptional cascade.

List of the new Figures

- Fig. 1b**: Over-expression of TASOR- Δ PARP mutant in addition of TASOR wt.
- Fig. 2c**: Interaction between endogenous TASOR and CNOT1.
- Fig. 2e**: The PARP domain of TASOR is required for its interaction with CNOT1.
- Fig. 3 (All new)**: TASOR and CNOT1 cooperate to repress HIV-1 expression in the J-Lat A1 model of HIV-1 latency.
- Fig. 4 (All new)**: TASOR and CNOT1 cooperate to repress HIV-1 expression in Latently HIV-1 infected Jurkat T cells
- Fig. 5 (All new)**: TASOR cooperates with CNOT1 to repress a subset of genes in HeLa cells. RNAseq analyses are available for reviewers on GEO, access number GSE184399, with token: arejawmqpvavdij
- Fig. 6b**: Endogenous TASOR interacts with the endogenous, nuclear CNOT1 and its partners CNOT7, YTHDF2, the endogenous TRAMP-like/NEXT/PAXT component MTR4, the endogenous RNA exosome factor EXOSC10.
- Fig. 7a and 7b**: TASOR colocalizes with active HIV-1 transcriptional centers in HIV-1 infected Jurkat cells (+ Fig S4a to S4e)
- Fig. 7d**: TASOR affinity for RNAPII is greatly reduced upon depletion of the N-terminal PARP domain of TASOR.
- Fig. 7g**: TASOR and CNOT1 (endogenous proteins) are found in complex with the HIV-1 nascent transcript.
- Fig. S1e**: Vpx R42A mutant is efficient to induce SAMHD1 degradation.
- Fig. S3c**: TASOR interacts with the nuclear TRAMP-like/PAXT complex member ZFC3H1.
- Fig. S4f**: modification to answer reviewer's comments

REVIEWER COMMENTS

Reviewer #1 (Remarks to the Author):

TASOR epigenetic repressor cooperates with a CNOT1 RNA degradation pathway to repress HIV
Matkovic et al.

In this study, the authors seek to identify new binding partners of the HUSH complex to gain some mechanistic insight into how the HUSH complex and associated complexes may repress HIV. In a yeast-2-hybrid screen, the authors identify CNOT1 as a novel interaction partner of the HUSH complex component TASOR. The HUSH complex exerts some homology with the yeast RNA-induced silencing

(RITS) complex, due to it possessing some of the same domains. Like the RITS complex, TASOR interacts with MTR4 and the exosome, which are binding partners of CNOT1. The RITS complex is associated with elongating RNA polymerase II and with small RNA-induced deposition of repressive H3K9me3 at retroelements. The interaction of TASOR with CNOT1, MTR4 and the nuclear exosome is dependent on RNA and TASOR interacts with elongating RNA pol II and the nascent HIV RNA reporter transcript. Interestingly, co-depletion of TASOR and CNOT1 or other associated RNA degradation factors leads to a synergistic increase in Luciferase production from the HIV reporter.

Depletion of DICER, in contrast, has no effect on HIV reporter transcripts. The authors conclude that HUSH represses HIV at the transcriptional and post-transcriptional levels through HUSH and the CNOT1 RNA degradation pathway. This study is novel, generally well-performed and interesting but some points need further clarification.

We thank the reviewer for his/her comments that help to improve our manuscript.

Major comments

1. Figure 1b: The TASOR-induced repression here is not remarkable (2-fold). It would be more convincing if this repression could be increased by increasing overexpression of TASOR. An additional negative control of GFP or other overexpressed construct should also be included.

TASOR-induced repression is indeed only 2-fold in our overexpression experiment. We suspect overexpression experiments to be difficult when several subunits are needed to form a functional complex like HUSH.

Nonetheless, we have added two controls (Fig 1b):

-Overexpression was carried out by adding increasing quantities of the vector expressing TASOR (only one dose the first time). Silencing is gradually enhanced.

-We have added a negative control: we have also overexpressed the TASOR- Δ PARP mutant, shown to be unable to repress gene expression in the study of Douse et al. 2020, Nature communications. The overexpression of TASOR- Δ PARP does not reduce LTR-driven gene expression, in contrast to TASOR wt. TASOR- Δ PARP even induces a slight increase of gene expression, perhaps due to the trapping of some host factors like MPP8 (Fig 2e)

Additionally, we now show that TASOR- Δ PARP does not bind CNOT1 or RNA polymerase 2 (Fig 2e and Fig 7d).

2. Figure 1cd: TASOR depletion does not affect total RNA (2.6-fold increase) much more than it affects nascent RNA (1.3-fold increase). In contrast, it appears to greatly affect protein levels (18x in figure 1c). This should be verified with RNA and protein measured in the same experiments and discussed (presumably TASOR tethers RNA to chromatin and its depletion causes RNA to be re-localized to the cytoplasm).

RNA and protein levels results previously shown were from the same experiment. To avoid confusion, they now appear under the same panel/letter (Fig 1c).

Of note, luciferase activity is deduced from an enzymatic assay and may not completely correlate with luciferase protein levels.

To answer the reviewer's concern, we have performed cell fractionation (next page) using the protocol published in *Nature protocols* by Gagnon et al. 2016 to recover Luc RNAs from the chromatin, nucleoplasmic and cytoplasmic fractions of HeLa LTR Δ TAR-Luc cells:

First, TASOR is not only present in the chromatin fraction but it is predominantly found in the nuclear soluble fraction as well as the HUSH members MPP8 and MORC2, which suggest that TASOR/HUSH has other functions than just binding to and propagating H3K9me3 marks. Second, siRNA-mediated TASOR depletion increases Luc RNA transcription from the HIV-1 LTR Δ TAR in these cells (Figs 1c; 2f) but, as suggested by Reviewer N°1, it indeed inhibits Luc RNA retention in the chromatin fraction and enhances its export towards the cytoplasm (Fig next page). The levels or distributions of the nucleus-localized MALAT1 lncRNA and ubiquitously displayed GAPDH RNA are not affected by TASOR depletion in these cells (next page). These results are part of another manuscript in preparation.

Overall and in agreement with the reviewer, the difference between RNA quantification and luciferase activity, which is much larger than the difference between nascent RNA and total RNA, could result from an impact of TASOR beyond RNA synthesis, on RNA export or even translation. This is now stated in the text ("the increase in luciferase activity after TASOR depletion was higher than the increase in Luc RNA levels when quantifying total RNA levels, suggesting that TASOR might impact steps beyond RNA turnover, such as RNA export or translation"). Consistent with this assertion, we identified members of the PAXT complex, which uses the exosome to degrade RNA, as interacting partners of TASOR: MTR4 and ZFC3H1 (ZFC3H1 being a new finding in this revised version). ZFC3H1 has been shown to functionally compete with nuclear export activity to retain target transcripts (Silla et al., 2018, Cell Reports) in agreement with our above comments.

HeLa LTRΔTAR-Luc

RNA quantification

RNA Distribution:

TASOR depletion:
 -increases Luc transcript synthesis (*Nuclear RUN ON*)
 -prevents Luc RNA retention on chromatin
 -increases Luc RNA cytoplasmic export

TASOR depletion:
 -increases MORC2 transcript synthesis (*Nuclear RUN ON*)
 -does not impact MORC2 export

TASOR depletion does not impact MALAT1 lncRNA levels or distribution

TASOR depletion does not impact GAPDH levels or distribution

3. It is not clear if TASOR and CNOT1 can inhibit the HIV replication cycle (exerting transcriptional and post-transcriptional blocks) and / or play a role in repressing latent HIV or if these complexes only act on transgenes, especially since TASOR seems to mainly affect chromatin accessibility across the Luciferase transgene (figure 4de). It would be interesting to have some insight into this question using WT HIV.

We have confirmed our results in two physiologically more relevant HIV models. One corresponds to a widely used HIV-1 model of latency using a repressed HIV-1 minigenome (J-Lat A1 cells as suggested by reviewer 2). TASOR was silenced using Vpx and CNOT1 using a miRNA encoding vector. The second model (LTR-EGFP model) has been developed following infection of the Jurkat T-cell line with a LTR-driven HIV-1 virus that has been developed by Yurkovetskiy et al., 2018 Nature Microbiology. The corresponding vector retains complete LTRs, *tat* and *rev*, but has a frameshift mutation in *env*, *ngfr* in place of *nef* and *egfp* in place of *gag*, *pol*, *vif* and *vpr*; the splicing sites are conserved. We have sorted infected cells by cytometry, thanks to EGFP expression and then, cells were left 2 weeks in culture until EGFP expression was silenced. Interestingly, while EGFP expression was not detected at the protein level in latently HIV-1 LTR-EGFP infected cells, HIV-1 unspliced RNAs were still detected (Fig 4b) suggesting a co/post-transcriptional silencing mechanism. Of note, in the J-Lat A1 model, a unique provirus is inserted in a unique position, while in the LTR-EGFP model, infected cells represent a population with diverse integration sites.

In both models, TASOR and CNOT1 were downregulated with Vpx and mirRNA, respectively. A synergistic repressive effect was observed between TASOR and CNOT1 in both systems (Fig 3 and Fig 4, entirely new).

In addition, we now present the results of an RNA-seq experiment to answer one concern about whether HUSH-CNOT1 could regulate host elements. We found a series of host genes cooperatively upregulated by the depletion of both TASOR and CNOT1, suggesting that this cooperation is not only dedicated to transgenes (Fig 5, entirely new). RNAseq bigWig processed data are available for reviewers on GEO, access number GSE184399, with token: arejawmqpvavdij

The question of latency. We have changed the discussion to discuss this point:

“Regarding HIV, our results with two models of HIV-1 latency may suggest HUSH to be involved in the maintenance of HIV latency. Nonetheless, the post-transcriptional activity of HUSH along with RNA synthesis suggests that HUSH activity needs the production of the viral RNA. However, we were unable to reveal a repressive role for HUSH in biological systems of high sustained viral RNA transcription, either due to proviral integration into transcriptionally active chromatin sites or due to high levels of the Tat protein. Since HUSH repression is dependent on the integration site¹⁰, it is conceivable that HUSH triggers H3K9 trimethylation - or its maintenance - only on provirus sequences integrated into poorly, but still, transcribing regions with signals of heterochromatinization. In this spatial window HUSH could counteract stochastic bursts of expression from the lowly expressed viral promoter⁶³. ”

Minor comments

1. Figure 1b: Lenti-DDK and Lenti-TASOR-DDK need to be defined in the legend. The TASOR band should be labelled as TASOR not DDK.

As suggested by the reviewer, we labelled the band corresponding to the exogenous TASOR as HA-TASOR or TASOR-DDK

2. Figure 1e: A Vpx mutant was used as a negative control that does not bind and induce degradation of TASOR. This control should be included in the figure rather than in supplementary data.

Following this suggestion, we moved this data from the supplementary data to Fig 1d.

3. Figure 2b: It would be clearer if 'DDK' was re-labelled as 'TASOR' on the IP or TASOR cDNA to distinguish from endogenous TASOR. This is true for many figures including figure 4fg.

In line with the reviewer's suggestion, we labelled the band corresponding to the exogenous TASOR as TASOR-DDK

4. In the first sentence of the abstract, LINE-1s, retrotransposons and retroelements are mentioned. This is confusing because LINE-1 elements are retrotransposons and retroelements.

The sentence has been changed to: The Human Silencing Hub (HUSH) complex constituted of TASOR, MPP8 and Periphilin is involved in the spreading of H3K9me3 repressive marks across genes and transgenes such as ZNF encoding genes, ribosomal DNAs, Retroelements, **including LINE-1 elements**, or the integrated HIV provirus¹⁻⁵.

5. Figure 2efg: Independent experiments are represented by different colours, which makes the figure unclear. It would be easier to represent different treatment groups by different colours.

6. Figure 3d: Can the authors comment on why there is so much variability in this figure?

We chose this presentation on purpose. Indeed, we wanted to highlight the reproducibility when comparing the different treatments within one experiment, rather than the reproducibility under a particular treatment. Therefore, we have not changed the colorization, but of course, we do agree to do it if the reviewer still prefers this alternative representation.

Variations in Luciferase expression from one experiment to the other under the same treatment may result from differences in cell passage or cell confluence. Indeed, we noticed that Luciferase expression decreased along with increasing cell passages. Although we cannot fully explain the reason, we believe that a transcriptional silencing mechanism that does not rely on HUSH occurs. Indeed, the downregulation of HUSH members did not fully restore Luc RNA expression as after the first passage event. However, we have a very good reproducibility when comparing the different treatments within the same experiment, which is the most important. In other word, for instance, the depletion of TASOR will always lead to an increase of luciferase activity, though the extent of this increase will vary from one experiment to the other. Similarly, synergy will be reproducibly obtained following depletion of both TASOR and CNOT1, compared to depletion of TASOR or CNOT1 alone.

7. Figure 4a: It would be helpful to show on this diagram that the MORC2 and TUG1 genes do not overlap. This could be shown by staggering the UCSC gene maps so that they are not simply a continuous line.

Accordingly, we have changed the MORC2-TUG1 view on this diagram which is now found in Fig S4f.

8. Figure 4g: I am not sure if it is necessary to have a split scale on the y axis. This graph would benefit from a simpler scale.

The corresponding graph has been removed because we have performed experiments with endogenous TASOR and CNOT1 proteins instead of exogenously expressed proteins, as recommended by Reviewer N°2 (Fig 7g).

9. Please revise the following unclear sentence: 'However, HIV-1 integration in poorly transcribed,

desert, geneless, or centromeric regions can result in the repression of HIV-1 expression due to multiple epigenetic mechanisms’.

The sentence has been changed by: “However, HIV proviral expression is also dampened by multiple epigenetic mechanisms, especially when integration occurs in poorly transcribed genes or geneless regions”

10. Please revise the following sentence as the word ‘repressive’ appears twice: ‘The HUSH complex - TASOR, MPP8 and periphilin- was identified as a potential player in HIV repression, propagating repressive H3K9 repressive marks in a position-effect variegation manner in a HIV-1 model of latency.’

One “repressive” word has been removed.

11. The text is unclear in parts and would benefit from some editing/re-writing and grammar corrections to aid readability.

We have corrected the text with more attention paid to grammar.

12. A structure prediction analysis of TASOR is mentioned: ‘Structure/function prediction analyses of TASOR first 900 amino acid (aa) sequence using the PSIPRED server (UCL Bioinformatics group) revealed a PARP13-like PARP domain at its N-terminus region (300 aa) with high probable and reliable functions in mRNA binding, splicing and processing, together with the SPOC (Spen paralog and ortholog C-terminal) domain, often associated with transcription repression’. Here, the authors should cite and refer to the recently published article (PMID: 33009411), in which structural details of TASOR, including its pseudo-PARP domain and the similarity between HUSH and the RITS complex are discussed.

We refer to the Douse et al paper much sooner now, when using the TASOR Δ PARP mutant in our overexpression experiment: “Notably, TASOR lacking its N-terminus PARP13-like PARP domain (Δ PARP), required for epigenetic repression according to Douse et al¹², does not repress LTR-driven expression in contrast to full-length TASOR (WT) (Fig. 1b).”

Later on, we now say: “Structure-function analysis provided additional hints on the potential mechanism of TASOR post-transcriptional effect. **Consistent with a recent study**¹², we found using the PSIPRED server (UCL bioinformatics group) that the N-terminus of TASOR is predicted to fold into a PARP13-like PARP domain with high probable and reliable functions in mRNA binding...”. The analogy with the RITS complex is discussed later on, with a reference again to the work of Douse et al. “The fission yeast RITS complex seems to match well our HUSH model, with yeast Chp1 and Tas3 sharing structural features with TASOR, MPP8 and PPHLN-1, **as also noticed by Douse et al**¹² and with the ability of RITS to interact with Not1/CNOT1⁵⁸, the exosome and MTR4 alike HUSH”

We keep saying that we have also seen these structural, and discovered these functional homologies, which we have prior to Douse et al. publication.

13. CNOT1 should be named in full at first mention and it would be helpful to state its cellular localization.

Full name has been added on L 87: “CNOT1 (CCR4-NOT transcription complex subunit 1)”.

We have modified the text as: “We chose to focus on CNOT1, which is the scaffold protein of the most important and conserved deadenylase complex from Yeast to Human, the Carbon catabolite repression 4-negative on TATA-less complex or CCR4-NOT^{27,28}.”

Later we have added: “By cell fractionation, we show that TASOR protein is predominantly detected in the nucleus, while CNOT1 is a shuttling protein, present both in the cytoplasm and nucleus, in agreement with previous studies⁴³⁻⁴⁵”

14. The authors should discuss why they couldn't efficiently overexpress CNOT1.

Actually, we do not know why we could not efficiently overexpress CNOT1. First, we received a construct from another lab, who told us they also had a lot of difficulty to reveal overexpressed CNOT1. We asked another colleague working with the Flag-CNOT1 construct who also struggled to reveal CNOT1. Then, we constructed our own expression vector, but without more success. We succeeded in revealing CNOT1 partners in a GFP-CNOT1 immunoprecipitate, however we could not reveal CNOT1 neither with an anti-CNOT1, nor with an anti-GFP. We suspect that overexpression of CNOT1 triggers large aggregates of the protein that is resistant to denaturation.

To be less confused, we have removed the sentence in which we were saying we could not efficiently overexpress CNOT1.

15. Please revise 'when inhibiting both proteins expression' to 'when depleting both proteins' or similar.

The change has been made.

16. Please revise the following unclear section: 'Because we could not efficiently overexpress CNOT1, we chose the deadenylase CNOT7 partner to pull-down the complex in the reverse situation. We immunoprecipitated the DDK-tagged CNOT7 and revealed an interaction with endogenous TASOR, and the other CCR4-NOT complex components CNOT9 and CNOT1 proteins (Fig. 2d)'.

We have simplified the section: “CNOT1 being a scaffold protein, we checked whether TASOR could interact with other components of the CCR4-NOT complex. By immunoprecipitating DDK-tagged CNOT7, the deadenylase partner of CNOT1, we revealed an interaction with endogenous TASOR, along with the other CCR4-NOT complex CNOT9 and CNOT1 components (Fig. S2e).”

17. All Westerns should have kDa size markers.

The size markers have been added to all Western-blot.

Reviewer #2 (Remarks to the Author):

Authors have previously reported in a very nice study in Nature Microbiology that TASOR is an important repressor of HIV expression, involving H3K9me3 histone mark, countered by the HIV-2 protein vpx. Matkovic et al. now describe a new function of TASOR in mRNA processing co/post-transcriptionally. Authors find new partners of TASOR which seem to cooperate in order to repress gene expression from an HIV-LTR/luciferase model. This complex is reminiscent of the yeast RITS complex. These findings are novel, in a well conducted study, and extend our understanding of how TASOR can repress expression of HIV at the co/post-transcriptional level. This study would however benefit from the following revisions.

We thank the reviewer for his/her comments which helped us to improve our manuscript.

Major points :

1) Authors should at some point use a better HIV model (full genome) to confirm that their new partners play an important role in a more physiological context. Although primary cells would be ideal, authors could also use the J-lat model they previously employed in their Nat. Microb.

We have confirmed our results in two physiologically more relevant HIV models. One corresponds to a widely used HIV-1 model of latency using a repressed HIV-1 minigenome (J-Lat A1 cells as suggested by reviewer 2). TASOR was silenced using Vpx and CNOT1 using a miRNA encoding vector. The second model (LTR-EGFP model) has been developed following infection of the Jurkat T-cell line with a LTR-driven HIV-1 virus that has been developed by Yurkovetskiy et al. (Nature microbiology, 2018). The corresponding vector retains complete LTRs, *tat* and *rev*, but has a frameshift mutation in *env*, *ngfr* in place of *nef* and *egfp* in place of *gag*, *pol*, *vif* and *vpr*; the splicing sites are conserved. We have sorted infected cells by cytometry, thanks to EGFP expression and then, cells were left 2 weeks in culture until EGFP expression was silenced. Interestingly, while EGFP expression was not detected at the protein level in latently HIV-1 LTR-EGFP infected cells, HIV-1 unspliced RNAs were still detected (Fig 4b) suggesting a co/post-transcriptional silencing mechanism. Of note, in the J-Lat A1 model, a unique provirus is inserted in a unique position, while in the LTR-EGFP model, infected cells represent a population with diverse integration sites.

In both models, TASOR and CNOT1 were downregulated with Vpx and mirRNA, respectively. A synergistic repressive effect was observed between TASOR and CNOT1 in both systems (Fig 3; Fig 4).

We regret we could not provide experiments with primary cells at this point. We hope the reviewer will agree that this is a very challenging issue. Indeed, as responded to Reviewer 1, we believe a low rate of transcription could be the key condition to highlight TASOR activity, in addition to position-effect variegation, i.e. integration of the provirus in a specific chromatin environment, as shown by Tchasovnikarova et al. 2015, Science. Up to now, we had to mimic HIV latency (with the J-Lat model or the LTR-EGFP model) to uncover HUSH activity.

2) IPs are not always sufficiently confirmed using endogenous proteins. Indeed, interactions are first assessed in Figure 2 using over-expressed proteins and both Y2H and over-expression experiments may yield false positives. However, authors later perform IPs using endogenous proteins that confirm some of their interactions. They should still confirm IP between TASOR and CNOT7 in their Figure 2 (either TASOR or CNOT7 IP would be sufficient). If this is not possible due to both antibodies not suitable for IP, authors could at least show WB of TASOR when TASOR-DDK is used and CNOT7 when CNOT7-DDK is used to compare them to endogenous levels (DDK alone). Same for figures 4b/c/f. There is no IP of endogenous proteins for phosphorylated forms of RNAPII. Also for RNAPII/TASOR interaction, there is endogenous IP in figure f, authors should show a stronger exposure of their blots to better assess endogenous TASOR presence after RNAPII IP ? Same for YTHDF2 and CNOT1 which seem absent unless there is overexpression. The fact that over-expression of the protein brings its associated partners is a good thing but endogenous proof is important.

We have confirmed the IP between endogenous TASOR and CNOT1/CNOT7 following an IP with the use of an anti-TASOR (Fig 2c and Fig 6b) and by performing immunoprecipitations with an anti-CNOT1 in HeLa cells (Fig 6c and Fig S2d). Interaction between endogenous TASOR and CNOT1 is also now provided in the J-Lat A1 T cell line (Fig 3a).

We also now show an interaction between endogenous TASOR and elongating RNA polymerase 2 (Ser2-P-RNA polymerase 2) and between endogenous TASOR and some RNA metabolism factors (Fig 6b).

We now used a stronger exposure for TASOR, CNOT1 and YTHDF2 signals when immunoprecipitating endogenous RNAPII (Fig 7f). We now better see the interactions between endogenous TASOR and

CNOT1 with RNAPII. However, the YTHDF2 signal exists but still remains low presumably due to indirect and weak interaction affinities.

3) An experiment of RNA-IP was performed at the end. This is a very nice experiment and suggests a direct effect of the TASOR. Authors could IP other proteins that interact with TASOR and probably function together to better confirm their model (e.g. at least one of the factors that have the best synergistic effect such as CNOT1). IPs against endogenous TASOR and/or some endogenous proteins (not only DDK-protein overexpression) are required (at least one protein of the complex should be studied using endogenous IP).

In agreement with our reviewer, we have now performed RNA-IP experiments against endogenous TASOR and endogenous CNOT1 (Fig 7g).

Minor points :

Line 124 : « between ... and » not « with »

The correction has been made (now on line 159)

Line 156 : I believe this experiment shows that both TASOR and CNOT complex can inhibit LTR driven luciferase mRNA, but do not formally demonstrate collaboration. At this point, we don't know if interaction is mandatory for the observed effect. I would change the phrasing accordingly (e.g. strongly suggest instead of confirm).

The correction has been made: "However, the inhibition of both CNOT7 and TASOR expression suggests a collaboration between CCR4-CNOT and TASOR (Fig. S2g)."

Fig 3d: CNOT7 fold increase seems to be missing. Does that mean it's not significant?

The fold increase disappeared from the graph by mistake when the figure was assembled. We thank the reviewer for their watchfulness. We have therefore added it in Fig 6e.

Reviewer #3 (Remarks to the Author):

Tasor epigenetic repressor cooperates with a CNOT1 RNA degradation pathway to repress HIV

Summary

In this study, Matkovic et al. propose that the Human Silencing Hub (HUSH) complex interacts with components of an RNA degradation pathway (CCR4-NOT) to repress HIV. The HUSH complex (TASOR, MPP8, and Periphilin) represses HIV provirus in addition to retroelements, retrotransposons and ribosomal DNAs through spreading of repressive epigenetic marks (H3K9me3). However, the precise mechanism of HUSH-mediated silencing is not clear. Using a protein-protein interaction approach, the authors found a physical interaction between TASOR and CNOT1, which is a scaffold of the RNA deadenylase CCR4-NOT complex. The authors then proposed that TASOR and CNOT1 interact in cells and "synergistically" repress HIV gene expression. In addition to CNOT1, TASOR appears to cooperate with MTR4 and the exosome, interacts with the RNA polymerase II elongating form and nascent transcripts, and TASOR over-expression facilitates the association of RNA polymerase II with the RNA degradation complex.

Overall, the study holds the potential for elucidating a previously undescribed functional link between

HUSH and an RNA degradation pathway in gene regulation. However, in its current format it is found preliminary and thus further evidence in physiologically relevant models of disease are required to substantiate the proposed models.

We thank the reviewer for his/her comments which helped to improve our manuscript.

Major concerns

1. In addition to the HIV provirus, HUSH regulates H3K9me3 spreading at human genomic domains including retrotransposons and retroelements. Providing evidence that HUSH–CCR4–NOT regulates these other host elements will broaden the scope of this work.

To determine whether our conclusion using HIV transcription models holds true for the different HUSH targets alike retroelements and endogenous repressed genes, we performed a transcriptomic analysis of cells depleted of TASOR, or CNOT1, or both. We highlighted a cooperation between TASOR and CNOT1 for a number of cellular genes, suggesting that our model could apply to different settings and does not only apply to transgenes.

Nonetheless, we were surprised we could not see an up-regulation of Line-1 RNAs following TASOR depletion, as we believe it should be the case on the basis of the literature. This is discussed in the discussion part: “To our surprise, we do not detect up-regulation of L1 RNAs following TASOR depletion, as expected from previous observations in other cell types^{2,4,11,12,36}, suggesting that additional and predominant repressive mechanisms may operate in HeLa cells to silence L1 elements, such as DNA methylation⁶². Alternatively, long-term shRNA depletion or full knock-out by CRISPR may be required to fully reveal the entire landscape of HUSH-regulated transcripts^{2,36}.”

2. Loss of TASOR triggers the accumulation of HIV transcripts in a HeLa latency model, and it is proposed that the effect is post-transcriptional rather than transcriptional. However, long-term TASOR depletion can lead to indirect consequences impacting on HIV gene expression, but this alternative interpretation was not carefully considered.

The reviewer is right to raise this point. We fully agree that long-term depletion of TASOR could have indirect consequences on HIV expression. Evidence for a direct role of TASOR is provided by the binding of TASOR to CNOT1 and RNA metabolism proteins (Figs 2b, 2c, 2d, 2e, 3a, 6b, 6c, 6d, S2d, S2e, S3a, S3c) the binding of TASOR to LTR-driven nascent transcript (Fig 7g), and the binding of TASOR to RNA polymerase 2 and mostly under its elongating state (Figs 7c, 7d, 7e, 7f). To reinforce these last two points, we now show by single molecule RNA FISH that TASOR colocalizes with HIV-1 transcriptional centers, where RNA polymerase 2 is synthesizing RNAs from the HIV-1 LTR, in Jurkat T CD4+ Cells (Figs 7a and 7b). We have also added the study of a TASOR mutant, which has no repressive effect in our system and does not bind to CNOT1 or RNA polymerase 2 unlike TASOR WT (Figs 2e and 7d).

3. In the abstract and throughout the text, the introduction that HIV-2 Vpx (which counteracts HUSH) mimics TASOR depletion is cumbersome and not well integrated in the entire story. It is known that Vpx induces HUSH degradation, but Vpx also has many accessory functions in the viral life cycle, thus it is unclear what events are HUSH-dependent and -independent Vpx functions. To assess this point, the authors could examine Vpx functions in the context of a relevant latency model with and without TASOR.

We hope the appearance of Vpx will be less cumbersome and better integrated in this new version. We have slightly changed the introduction of Vpx and we have used it more often throughout the different experiments (specially to test the synergy between TASOR and CNOT1 in two models of HIV-1 latency). Though, it is true that Vpx is rather used as a tool to study HUSH restriction toward HIV-1 than to study its own activity in the context of HIV-2. We have added a paragraph in the discussion to further discuss the role of Vpx.

Of note, the Vpx mutant, which has lost the ability to increase LTR-driven transcript stability, has kept the ability to induce SAMHD1 degradation, an important and well-known function of Vpx, suggesting that the defect is not linked to SAMHD1 at least. We have added this new information (Fig S1e).

4. Much of the work builds on a non-physiological relevant model in HeLa cells containing an LTR-driven luciferase reporter in which the TAR stem-loop has been deleted (Figure 1). This molecular event does not occur in replication-competent proviruses, but only under special circumstances in proviruses that undergo latency (e.g., work by Verdin lab). Also, HeLa are not relevant to HIV infection and the observed phenotype by which TAR deletion enhances HIV gene expression is artificial. Physiologically relevant systems (e.g., CD4+ T cells) must be put into work to define how HUSH could inhibit the turnover of HIV transcripts.

We have now confirmed our results with two physiologically more relevant HIV models. One corresponds to a widely used HIV-1 model of latency using a repressed HIV-1 minigenome (J-Lat A1 cells, from Eric Verdin, as suggested by reviewer 2). TASOR was silenced using Vpx and CNOT1 using a miRNA encoding vector. The second model (LTR-EGFP model) has been developed following infection of the Jurkat T-cell line with a LTR-driven HIV-1 virus that has been developed by Yurkovetskiy et al. (Nature microbiology, 2018). The corresponding vector retains complete LTRs, *tat* and *rev*, but has a frameshift mutation in *env*, *ngfr* in place of *nef* and *egfp* in place of *gag*, *pol*, *vif* and *vpr*; the splicing sites are conserved. We have sorted infected cells by cytometry, thanks to EGFP expression and then, cells were left 2 weeks in culture until EGFP expression was silenced. Interestingly, while EGFP expression was not detected at the protein level in latently HIV-1 LTR-EGFP infected cells, HIV-1 unspliced RNAs were still detected (Fig 4b) suggesting a co/post-transcriptional silencing mechanism. Of note, in the J-Lat A1 model, a unique provirus is inserted in a unique position, while in the LTR-EGFP model, infected cells represent a population with diverse integration sites.

In both models, TASOR and CNOT1 were downregulated with Vpx and mirRNA, respectively. A synergistic repressive effect was observed between TASOR and CNOT1 in both systems (Fig 3; Fig 4).

We regret we could not provide experiments with primary cells at this point. We hope the reviewer will agree that this is a very challenging issue. Indeed, as responded to Reviewer 1, we believe a low rate of transcription could be the key condition to highlight TASOR activity, in addition to position-effect variegation, i.e. integration of the provirus in a specific chromatin environment, as shown by Tchasovnikarova et al. 2015, Science. Up to now, we had to mimic HIV latency (with the J-Lat model or the LTR-EGFP model) to uncover HUSH activity.

5. Throughout the text, overexpression of proteins has been used to propose models of HIV gene regulation (Figure 1), but it is unclear what is the physiologic relevance of these artificial experiment in HeLa cells.

We do agree that overexpression experiments as the one shown in Figure 1 are somehow artificial, and the interpretation might be difficult. In this new version, we also overexpressed the TASOR- Δ PARP mutant, shown to be inactive in the study of Douse et al¹² (no silencing). The overexpression of TASOR- Δ PARP does not reduce gene expression, in contrast to TASOR WT. TASOR- Δ PARP even induces a slight increase of gene expression, perhaps due to the trapping of some host factors alike MPP8 (Fig 2e). Additionally, we now show that TASOR- Δ PARP does not interact with CNOT1 or RNA polymerase 2 (Figs 2e and 7d).

6. An interaction between HUSH and CCR4-NOT must be shown in physiologically relevant systems and not under overexpression conditions (Figure 2).

We have confirmed the IP between endogenous TASOR and CNOT1/CNOT7 following an IP with the use of an anti-TASOR (Fig 2c and Fig 6b) and by performing immunoprecipitations with an anti-CNOT1 in HeLa cells (Fig 6c and Fig S2d). Interaction between endogenous TASOR and CNOT1 is also now provided in the J-Lat A1 T cell line (Fig 3a).

We also now show an interaction between endogenous TASOR and elongating RNA polymerase 2 (Ser2-P-RNA polymerase 2) and between endogenous TASOR and some RNA metabolism factors (Fig 6b).

7. The authors utilized publicly available CHIP-seq and RNA-seq datasets (Liu et al.) to assess factor and epigenetic mark occupancy, but it was unclear in the text in what cell types these were collected and whether they hold physiologic relevance.

CHIP-seq and RNA-seq datasets of Liu *et al* were performed in human chronic myeloid leukemia K562 cells, which do not have physiological relevance for HIV. Nonetheless, the analysis of these results gave us an important clue for investigating further an interaction between HUSH, RNA polymerase 2 and nascent RNA.

Minor concerns

1. The text needs grammar and scientific improvement. Example on pg. 3. Vpx does not “degrade” TASOR-MPP8 because it is not an E3 ligase itself, but it is needed for “TASOR-MPP8” degradation. Likewise, TNF is not a “transcriptional activator” per se, but a known ligand that induces a signaling and transcriptional cascade.

Accordingly, we have corrected these mistakes

Reviewers' Comments:

Reviewer #1:

Remarks to the Author:

The authors have addressed all of my concerns in this revised ms thank you. Of note, there is an additional figure (Fig 5) showing several retroelements (ERV1 & ERV9-LTR12) repressed by TASOR/CNOT1. These two are also the most derepressed LTRs upon MPP8-inactivation in a recent study (PMID: 33144593). This is worth mentioning to help to build an overview of which retroelements HUSH regulates.

Reviewer #2:

Remarks to the Author:

Reviewer's comments were addressed. I thank the authors for their work. The manuscript is now improved and better characterizes the cooperation between TASOR and CNOT1.

Reviewer #3:

Remarks to the Author:

The revised manuscript is much improved, and the authors adequately responded to prior requests (e.g., expanded studies to models of HIV-1 latency, included immunoprecipitations with endogenous proteins, RNA-seq to examine transcriptome changes upon TASOR silencing, etc.).

Major comments:

1. Most if not all experiments in this manuscript utilize siRNA or miRNA-mediated factor silencing. Thus, a limitation of the study that must be highlighted is that indirect effects cannot be ruled out and could have a large impact on the results obtained.
2. A fundamental unanswered question is why HUSH must recruit RNA degradation machinery to enforce HIV silencing? Is HUSH-mediated epigenetic silencing ineffective on its own since the main effects appears to be post-transcriptional?

Minor comments:

1. Sentences describing increase or decrease in activity or interactions may benefit from providing quantitative data (fold change) and statistics to help the reader.
2. I did not find information regarding statistics in the figure legends and figures do not seem to have statistical tests beyond showing error bars.
3. Does HUSH trigger H3K9 methylation directly or does it recruit H3K9 methylases (SETDB1)? Sentences describing this concept should be revised for more clarity.
4. Can the authors better explain why HUSH may act on HIV-1 provirus integrated into specific sites but not others? Is it recruited to some sites but not others? Is the downstream CNOT-mediated effect on HIV RNA degradation integration-site dependent too?
5. Line 27 (abstract), it should be more clearly stated that HUSH silences the HIV provirus in an integration site-dependent manner.
6. Line 70 should read "deposition"
7. Lines 106-107. "To examine the role of TASOR on mRNA metabolism...." Run on measures transcription per se.
8. Line 110, "very slightly increases". Is this 1.3-fold increase statistically significant?
9. Are the results of Figure 3c a consequence of Tat activity and how does this data relate to data in Figure 1a which uses a delta-TAR LTR model?
10. Some sentences (e.g., lines 227-228) will guide the reader better if conclusions are expanded. What does it mean that "such synergy was not observed on MORC2 or TNF genes"?
11. Line 281. "depleted" and other places throughout the main text. This should be "silenced" because

RNAi is used.

12. References 5 and 10 are the same one.

13. Figure 2a will benefit from incorporating quantitative data and statistics in addition to showing which factors visually interact with TASOR.

14. Figure 3. Why does this figure use miRNA rather than siRNA or shRNA for CNOT1 silencing? Are the silencing effects reversible? Have the authors tried complementing with a CNOT1 construct to demonstrate the effect is on target?

Reviewers #1 and #2

We are glad the reviewers are satisfied by the revised version of our manuscript.

We now mentioned the manuscript highlighted by Reviewer 1 showing that two retroelements derepressed in our TASOR/CNOT1 conditions are the most derepressed LTRs upon MPP8 inactivation (PMID: 33144593).

“Of note LTR12C and HERV9NC-int represent the LTR and internal sequence of the same element, a variant of the HERV-9 family, respectively and ERV9-LTR12 was already shown to be the most derepressed LTR upon MPP8-inactivation (Tunbak et al., 2020)”.

Reviewer #3

We thank the reviewer for his comments, which greatly helped us to improve our manuscript. We are glad he found our manuscript much improved, with adequate responses to prior requests.

To answer the new comments, we have introduced modifications all along the text, taking into consideration carefully the reviewer’s comments. For reasons explained below, complementation experiments seem very uncertain at this stage. We hope the reviewer will be satisfied with this new version.

Major comments:

1. Most if not all experiments in this manuscript utilize siRNA or miRNA-mediated factor silencing. Thus, a limitation of the study that must be highlighted is that indirect effects cannot be ruled out and could have a large impact on the results obtained.

The reviewer is right, we may also observe indirect effects because of the use of siRNA or miRNA.

For TASOR siRNA, we would like to argue that we had similar effects by using Vpx wt, and not a Vpx point mutant unable to degrade TASOR, both in relevant HIV models (Figures 3 and 4).

For CNOT1 siRNA, we would like to argue that (i) we have similar effects using CNOT1 or CNOT7 siRNA; it is unlikely that same off-targets are at stake in the resulting effects.

In addition, our study reveals an interaction between TASOR N-terminal domain and CNOT1 (interaction and mapping by the Y2H assay, and interactions confirmed by co-immunoprecipitations). The TASOR Δ PARP protein that is unable to interact with CNOT1, is not able to repress the HIV-1 LTR expression. We have also confirmed the interactions between TASOR and other RNA metabolism factors.

Furthermore, for siRNA or miRNA-mediated silencing of TASOR and CNOT1 expression, we likely also observed direct effects as demonstrated by the Nascent RNA IP showing that both TASOR and CNOT1 are found in association with the LTR-derived transcript. In line with this, we have modified the text “Furthermore, after BrUTP labeling of nascent transcripts, we coupled immunoprecipitation of endogenous TASOR or CNOT1 to RT-qPCR and found that the nascent LTR-derived Luc transcript was equivalently associated with TASOR and CNOT1, which underlines their direct functions on HIV-1 LTR-derived transcript”.

To better take into consideration the reviewer’s comment, we have added in the discussion the following sentence (highlighted in green):

“Overall, our results support a model in which TASOR, in association with CNOT1, provides a platform along transcription to destabilize nascent transcripts. This conclusion is supported by:(viii) the

accumulation of TASOR in transcription centers and the interaction of TASOR and CNOT1 with nascent transcripts. While indirect effects cannot totally be ruled out from siRNA, miRNA or Vpx-mediated deletion experiments, this latter point suggests a direct effect of TASOR and CNOT1 on LTR-derived RNA metabolism.”

2. A fundamental unanswered question is why HUSH must recruit RNA degradation machinery to enforce HIV silencing? Is HUSH-mediated epigenetic silencing ineffective on its own since the main effects appears to be post-transcriptional?

Our results highlight a new role for HUSH, which is to silence HIV-1 expression at a co- and post-transcriptional stage. TASOR actually seems to be part of an RNA surveillance system. The mechanism of HUSH recruitment is still enigmatic. Our results suggest that the silencing mechanism of HUSH needs the presence of an active transcription process with the synthesis of an RNA: TASOR binds the elongating RNA polymerase 2, is located in HIV-1 active transcriptional centers, and is found in association with the nascent RNA and recruits RNA degradation factors. The PARP domain-truncated version of TASOR is not able to bind the deadenylase complex component CNOT1 and is not able to repress HIV-1 expression (Fig 1B) nor to induce epigenetic silencing (Douse *et al.*, Nature Communications 2020, PMID: 33009411).

As discussed in our manuscript, a recent paper showed that MPP8 truncated of its chromodomain (necessary for H3K9me3 binding and spreading) is still able to repress LINE-1 elements (Muller *et al.*, Nature Communications 2021, PMID: 34031396), suggesting, keeping our results in mind, that MPP8 could also exert a post-transcriptional activity in this system. It is not known whether the reverse (repression at the epigenetic level unrelated to a post-transcriptional pathway) would be possible. Then, a paper showed that the other HUSH member, Periphilin-1 (necessary for H3K9me3 deposition as well – Prigozhin *et al.*, Nucleic Acids Research, 2020, PMID: 32976585), is found in association with RNAs (Castello *et al.*, Cell 2012, PMID: 22658674). Thus, HUSH represses retroelements at an epigenetic/transcriptional level and perhaps due to recognition of the target nascent RNA in a specific gene environment. Why would HUSH recruit RNA degradation machinery? Perhaps to prevent the expression of genes, retroelements from stochastic bursts of transcription and prevent the expression of “abnormal” RNAs that could lead to a pathogenic phenotype? Thus, yes, the reviewer is right, HUSH at the epigenetic level might be ineffective on its own. This is still a mystery and knowing how and why HUSH is recruited might answer this question. In any case, we and the group of Yorgo Modis have found homologies with the yeast RITS complex (Douse *et al.*, Nature Communications 2020, PMID: 33009411). The latter also recruits the TRAMP complex to degrade the nascent RNA.

To better take into consideration the reviewer’s comment, we have added in the discussion the following sentence (highlighted in green):

.....suggesting that in this configuration, HUSH could repress its targets independently from chromatin binding and maintenance of H3K9me3. It is not yet known whether the reverse, i.e. repression at the epigenetic level unrelated to RNA degradation mechanisms, is also possible. Therefore, a fundamental unanswered question is whether HUSH-mediated epigenetic repression could be effective by itself or whether HUSH must always rely on RNA degradation factors recruitment to enforce HIV silencing.

Minor comments:

1. Sentences describing increase or decrease in activity or interactions may benefit from providing quantitative data (fold change) and statistics to help the reader.

We often provide quantitative data when describing results throughout the manuscript but accordingly we have added more details in the text now.

2. I did not find information regarding statistics in the figure legends and figures do not seem to have statistical tests beyond showing error bars.

The legends indeed state whether SEM or SD is shown, the number of replicates, the statistical tests and their results (e.g : lines: **974-975; 995-996; 1007-1008; 1013-1014; 1016-1018; 1050-1051; 1075-1076; 1144-1155; 1166-1167**). As previously answered the reviewer 1, when it comes to perform statistical analysis regarding the luciferase measurements we face an obstacle: Variations in luciferase expression from one experiment to the other under the same treatment may result from differences in cell passage or cell confluence. Indeed, we noticed that Luciferase expression decreased along with increasing cell passages. Although we cannot fully explain the reason, we believe that a transcriptional silencing mechanism that does not rely on HUSH occurs. We then chose to plot each value on a graph to show how reproducible the treatment is, with the corresponding fold change value.

3. Does HUSH trigger H3K9 methylation directly or does it recruit H3K9 methylases (SETDB1)? Sentences describing this concept should be revised for more clarity.

Our reviewer is right, we should have specified the role of SETDB1 in the H3K9me3 spreading. We have modified the first sentence of the abstract to “The Human Silencing Hub (HUSH) complex constituted of TASOR, MPP8 and Periphilin **recruits the histone methyl-transferase SETDB1** to spread H3K9me3 repressive marks across genes and transgenes such as ZNF encoding genes, ribosomal DNAs, retroelements, **including LINE-1 elements**, or the HIV **provirus in an integration site-dependent manner**¹⁻¹⁰”.

We also have modified (L.62-66) “The HUSH complex – formed by TASOR (*alias* C3orf63, FAM208A), MPP8 and PPHLN-1 - was identified as a potential player in HIV repression, propagating repressive H3K9me3 marks **with the help of the histone methyl-transferase SETDB1** and resulting in position-effect variegation in an HIV-1 model of latency^{1,10}”

4. Can the authors better explain why HUSH may act on HIV-1 provirus integrated into specific sites but not others? Is it recruited to some sites but not others? Is the downstream CNOT-mediated effect on HIV RNA degradation integration-site dependent too?

As we discussed in our manuscript, we were unable to reveal a repressive role for HUSH in biological systems of high sustained viral RNA transcription, either due to proviral integration into transcriptionally active chromatin sites or due to high levels of the Tat protein (When using HeLa-LAV cells that harbor highly transcribing copies of HIV-1, we did not observe any effect of CNOT1 silencing, nor of TASOR silencing, at the RNA levels). Since HUSH repression is dependent on the integration site (Tchasovnikarova *et al.*, 2015, PMID: 26022416) it is conceivable that HUSH triggers H3K9 trimethylation - or its maintenance - only on provirus sequences integrated into poorly, but still, transcribing regions with signals of heterochromatinization. In this spatial window, HUSH could counteract stochastic bursts of expression from the lowly expressed viral promoter. As of today, the exact mechanism that triggers the recruitment of HUSH onto its target remains poorly understood.

Whether the downstream CNOT1-mediated effect is also integration-site dependent is a good question. We would expect CNOT1 effect to be integration-dependent when CNOT1 is working with HUSH. However, it is likely that CNOT1 has a role in destabilizing transcripts independently from HUSH, due to its central role in promoting mRNA deadenylation in eukaryotic cells, and in this case its impact could be independent from proviral integration. Supporting this hypothesis, we have found that the depletion of CNOT1 increased proviral expression in the J-Lat A2 cells, which do not respond to Vpx addition. In this case, expression of the integrated provirus seems dependent on CNOT1, but not on HUSH.

5. Line 27 (abstract), it should be more clearly stated that HUSH silences the HIV provirus in an integration site-dependent manner.

The reviewer is right to correct us. Accordingly, we have modified this sentence by “The Human Silencing Hub (HUSH) complex constituted of TASOR, MPP8 and Periphilin recruits the histone methyltransferase SETDB1 to spread H3K9me3 repressive marks across genes and transgenes such as ZNF encoding genes, ribosomal DNAs, retroelements, including LINE-1 elements, or the HIV provirus in an integration site-dependent manner¹⁻⁵”

6. Line 70 should read “deposition”

In agreement with the reviewer we have replaced all “deposit” terms by “deposition”.

7. Lines 106-107. “To examine the role of TASOR on mRNA metabolism....” Run on measures transcription per se.

The author is right when they say that the *Nuclear Run On (NRO)* technique enables the measurement of Transcription *per se*. We have changed our sentence to be clearer: “To examine the role of TASOR in mRNA metabolism, we performed *Nuclear Run On* experiments to assess the effect of TASOR silencing on nascent Luc RNA transcription from the HIV-1 LTR, while quantifying this transcript at the total level. Then, by comparing nascent Luc transcripts (labeled with BrUTP) to the total Luc mRNA amounts (Fig. S1a), we can determine at which stage of RNA metabolism TASOR may negatively act.”

8. Line 110, “very slightly increases”. Is this 1.3-fold increase statistically significant?

We have performed three Nuclear Run On experiments:

We always observe this slight increase in Luc Nascent RNA levels. Using the unpaired t-student test, the p-value is at the limit of significance: $p = 0.0552$. We then preferred writing “While TASOR down-regulation by siRNA **very slightly increases** the levels of nascent transcripts (1.3-fold, Fig. 1c)”

9. Are the results of Figure 3c a consequence of Tat activity and how does this data relate to data in Figure 1a which uses a delta-TAR LTR model?

The delta-TAR LTR model helped us to unravel the synergistic effect between TASOR and CNOT1 and to dissect the underlying mechanism. The objective of the use of the J-Lat A1 Model and the LTR-GFP model in Figures 3 and 4 was to validate our results in more relevant models for HIV, as requested by the reviewers. But, of course, the reviewer is right, these models are very different. As both the delta-TAR model and the J-Lat A1 model are sensitive to HUSH, we do not believe that the presence of Tat makes the difference to uncover HUSH activity. Of note, when overexpressing Tat in a HeLa LTR-luc

model (high transcription rate), we could not uncover any effect of HUSH. Altogether, we suspect the rate of transcription to be an important parameter to reveal HUSH activity. To answer the reviewer's comment, it would be nice to develop a LTR-driven transcription model, in which Tat expression could be induced at different levels and, in parallel, to study the sensitivity to HUSH (by adding Vpx).

10. Some sentences (e.g., lines 227-228) will guide the reader better if conclusions are expanded. What does it mean that "such synergy was not observed on MORC2 or TNF genes"?

We are happy to clarify this sentence. MORC2 and TNF genes are not co-regulated by TASOR and CNOT1 as the integrated HIV-1 provirus in the CD4+ lymphocytic cell line.

We have changed the text: Analysis of EGFP expression by cytometry or quantification of unspliced RNA transcribed from the 5'LTR demonstrate the synergy between TASOR and CNOT1 in repressing HIV-1 LTR-driven expression in T cells (Fig. 4d and 4e, 16.1x fold increase when silencing both TASOR and CNOT1 expressions in comparison to 3.0x and 3.6x upon TASOR and CNOT1 silencing respectively). Only a 2-fold increase in MORC2 transcript levels (corresponding to TASOR silencing) and no effect on the TNF α transcripts were observed under these experimental conditions (Fig. 4f and 4g).

11. Line 281. "depleted" and other places throughout the main text. This should be "silenced" because RNAi is used.

In line with our reviewer's remark, we have replaced "depleted" or "depletion" by "silenced" or "silencing".

12. References 5 and 10 are the same one.

We thank our reviewer for their watchfulness. References have been corrected.

13. Figure 2a will benefit from incorporating quantitative data and statistics in addition to showing which factors visually interact with TASOR.

The Y2H screen was performed only once. In Fig S2C we have given a list of TASOR 1-900 interacting proteins that were identified with this experiment. As indicated in the legend of this panel S2C, interacting partners were assigned a predicted biological score from A-F to assess the confidence of an interaction being specific (with A indicating very high confidence, and F indicating experimentally determined artifacts). The interactions in solid blue lines are new interactors that we have confirmed by co-immunoprecipitation by at least 3 experiments and some of them are presented in the manuscript as stated in the legend.

14. Figure 3. Why does this figure use miRNA rather than siRNA or shRNA for CNOT1 silencing? Are the silencing effects reversible? Have the authors tried complementing with a CNOT1 construct to demonstrate the effect is on target?

This construct was kindly given by Dr. Alfonso Rodriguez-Gil who used it in his Scientific Report paper in 2017 (PMID: 28615693). In this paper, he has validated the silencing of CNOT1 using this pINDUCER10 and upon doxycycline treatment. This construct was then suitable for us since we struggle in transfecting Jurkat cells with siRNAs and to avoid long-term depletion of CNOT1 with shRNAs that could lead indeed to indirect effects due to its pivotal role in gene expression. With VLP-mediated

pINDUCER10 miR-CNOT1 transduction and puromycin-mediated selection of cells that have integrated this construction, we have generated mirCNOT1 J-Lat A1 and HIV-1 latently infected Jurkat cells. After 72h of Doxycycline treatment to inhibit transiently CNOT1 expression, the majority of cells became GFP positive. When we stopped Dox-treatment, we observed that the cells became GFP negative in one week, so this is reversible.

Complementation experiments is a great idea. However, we face two major obstacles:

-As previously responded to reviewer N°1, we could not efficiently overexpress this protein. First, we received a construct from another lab, who told us they also had great difficulties revealing overexpressed CNOT1. We asked another colleague working with the Flag-CNOT1 construct who also had trouble expressing/revealing CNOT1. We then constructed our own expression vector, but without further success. We suspect that overexpression of CNOT1 triggers large aggregates of the protein that resist denaturation and that cannot then fulfill its function. Complementation experiments with CNOT1 seem very uncertain under these conditions.

-Furthermore, due to the central role of CNOT1 in mRNA metabolism (from synthesis, through export and to RNA degradation), overexpression above physiological/normal levels would, unfortunately, not give the expected phenotype.